# Stabilizing atomic Ru species in conjugated sp² carbon-linked covalent organic framework for acidic water oxidation

Hongnan Jia[1,3], Na Yao[2,3], Yiming Jin[1], Liqing Wu[1], Juan Zhu[1] & Wei Luo ✪[1] ✉

Suppressing the kinetically favorable lattice oxygen oxidation mechanism pathway and triggering the adsorbate evolution mechanism pathway at the expense of activity are the state-of-the-art strategies for Ru-based electro-catalysts toward acidic water oxidation. Herein, atomically dispersed Ru species are anchored into an acidic stable vinyl-linked 2D covalent organic framework with unique crossed π-conjugation, termed as COF-205-Ru. The crossed π-conjugated structure of COF-205-Ru not only suppresses the dissolution of Ru through strong Ru-N motifs, but also reduces the oxidation state of Ru by multiple π-conjugations, thereby activating the oxygen coordinated to Ru and stabilizing the oxygen vacancies during oxygen evolution process. Experimental results including X-ray absorption spectroscopy, in situ Raman spectroscopy, in situ powder X-ray diffraction patterns, and theoretical calculations unveil the activated oxygen with elevated energy level of O 2$p$ band, decreased oxygen vacancy formation energy, promoted electrochemical stability, and significantly reduced energy barrier of potential determining step for acidic water oxidation. Consequently, the obtained COF-205-Ru displays a high mass activity with 2659.3 A g$^{-1}$, which is 32-fold higher than the commercial RuO$_2$, and retains long-term durability of over 280 h. This work provides a strategy to simultaneously promote the stability and activity of Ru-based catalysts for acidic water oxidation.

The proton exchange membrane water electrolyzer (PEMWE) provides a prospective technology for industrial hydrogen production owing to its advantages of high current densities, satisfactory hydrogen purities and low energy losses[1–3]. Unfortunately, the large-scale application of PEMWEs is highly restricted by the sluggish anodic oxygen evolution reaction (OER). Currently, scarce iridium oxide (IrO$_2$) has been considered as the state-of-the-art OER catalyst under the harshly acidic electrolyte, due to the optimal balance between activity and stability[4–7]. In general, IrO$_2$ follows the conventional adsorbate evolution mechanism (AEM) pathway[8,9], which involves multiple oxygen-containing intermediates with linear scaling relation, thereby leading

to unsatisfactory OER kinetics. Recently, ruthenium oxide (RuO$_2$) is regarded as a promising alternative to IrO$_2$ due to its low cost and high intrinsic activity[10,11]. More critically, unlike IrO$_2$, the lattice oxygen in RuO$_2$ could participate in the OER through the lattice oxygen oxidation mechanism (LOM) pathway, breaking the scaling relation between the adsorption energy of *OOH and *O in the AEM pathway, resulting in significantly enhanced intrinsic OER kinetics[12–14]. Consequently, it is expected that activating the lattice oxygen and reducing the oxygen vacancy formation energy of RuO$_2$ during the OER process could further enhance the activity, but has been rarely reported.

[1]College of Chemistry and Molecular Sciences, Wuhan University, Wuhan, Hubei 430072, PR China. [2]State Key Laboratory of New Textile Materials and Advanced Processing Technologies, Wuhan Textile University, Wuhan, Hubei 430073, PR China. [3]These authors contributed equally: Hongnan Jia, Na Yao. ✉e-mail: wluo@whu.edu.cn

Although LOM pathway endows the elevated water oxidation kinetics of $RuO_2$, the generation of lattice oxygen vacancies and soluble high-valent $RuO_4^{2-}$ species usually result in collapse of the crystal structure of $RuO_2$ and degraded stability[11,12,15]. To this end, recent studies focus on regulating the electronic structure of $RuO_2$ to trigger the AEM pathway and inhibit the LOM pathway to increase stability at the expense of intrinsic activity[11,16,17]. Therefore, simultaneously promoting the catalytic activity and stability of Ru-based catalysts by conventional methods still remain challenging. In fact, from a structural viewpoint, precisely constructing a strong metal–support interaction to fix atomically disperad Ru units is a promising strategy to stabilize soluble active species[18–20]. Noteworthy, previous studies show that π-conjugated system can ensure effective charge delocalization to construct an efficient electronic path, further strengthening the metal–support interaction[21]. Specifically, two-dimensional (2D) covalent organic frameworks (COFs) are known for their highly accessible surface area, good electrical conductivity and well-defined structure, which combined with their customizable pore environments, make them extensively employed as an effective platform to anchor active metal centers for improved catalytic performances in the field of energy and environment[22], including photocatalysis[23–25], $CO_2$ reduction[26,27], alkaline water splitting[28,29] and so on. Sun et al. reported an O–O coupling process in an Aza-fused, π-conjugated, microporous polymer coordinated single cobalt sites (Aza-CMP-Co) during alkaline OER process, which proves the feasibility of the intramolecular O–O coupling mechanism by single atomic sites in COF structure[20]. Unfortunately, when the electrolyte turns to acidic condition, traditional COFs with conventional imine linkages usually suffer from the collapse of structure[30–32]. Therefore, it is critical to develop advanced acidic-stable COF with effective π-conjugation and further anchor Ru complexes to deliver both enhanced activity and stability toward acidic OER.

Herein, we design and construct an acidic-stable COF-based electrocatalyst by incorporating atomic Ru species into a vinyl-linked 2D COF with unique crossed π-conjugation (COF-205-Ru). We find the crossed π-conjugated systems with multiple π-conjugations show enhanced electron orbitals overlap, which is responsible for the optimized electronic structure and promoted electron transfer efficiency. We unravel that the precise structure of COF-205-Ru not only suppresses the dissolution of Ru through a strong Ru–N bond, but also reduces the oxidation state of Ru by crossed π-conjugation, leading to stabilized oxygen vacancies derived from the coordinated oxygen departure. Specifically, the obtained COF-205-Ru displays a high mass activity (2659.3 A g$^{-1}$) and long-term durability (over 280 h) toward acidic OER. The experimental results containing X-ray absorption spectroscopy (XAS) analysis, in situ Raman spectroscopy and in situ powder X-ray diffraction (PXRD) patterns unveil the activation and participation of coordinated oxygen during OER processes with high stability. Theoretical calculations indicate that the unique crossed conjugation of COF-205-Ru can not only provide strong electron delocalization, stabilize the oxygen vacancies, and lower the energy barrier of the rate-determining step (RDS), but also maintain the structure stability by robust π-conjugated framework and Ru–N motifs. Since this is the first example of the application of COFs toward acidic OER and exploration of the structure–property relationship between the active site structure and its basic catalytic activity/stability would provide a new research paradigm for the precise design of acidic OER catalysts and fundamental identification of active centers in terms of catalyst structures.

## Results

### Principle of crossed π-conjugation for COF-205-Ru

Taking advantage of the strong chelating ability of bipyridine, we designed the 2D COF-205 structure by integrating 2,2'-bipyridine-5,5'-dialdehyde (Bpy-DA) as a building block. Interestingly, compared to $RuO_2$, the delocalized crossed π-conjugation systems are constructed in COF-205-Ru by the (i) π–π conjugation of 2D COF-205-Ru frameworks, (ii) strong $d$–π hybridization between π orbitals of COF-205 and unoccupied $d_{xz}/d_{yz}$ orbitals of Ru and (iii) $d$–$p$ interaction between $p$-orbital of oxygen and unoccupied $d_{xz}/d_{yz}$ orbitals of Ru (Fig. 1a). Importantly, these π-conjugated systems lead to a significant charge delocalization, providing favorable electron paths. According to the charge density difference analysis, strong electronic delocalization is observed in COF-205-Ru, indicating the electrons flow to Ru sites through the π-conjugated motifs (Fig. 1b). Benefiting from the electronic structure optimization on Ru sites, it is expected that the electropositive oxygen vacancies generated during the OER process will be stabilized. Moreover, the unique crossed π-conjugation systems elevate the energy level of O $2p$ band, thereby leading to a much higher area of the non-bonding oxygen ($O_{NB}$), and activation of coordinated oxygen (Fig. 1c). In addition, the N $2p$ band passes through the Ru $4d$ band, indicating that the formation of strong Ru–N interaction, which is conducive to increasing the stability during the catalytic process. Consequently, achieving the 2D COF anchored Ru with crossed π-conjugation is essential for the simultaneously enhanced electrocatalytic activity and durability toward acidic OER. Specifically, the designed COF-205 network with robust $sp^2$-carbon linkage was prepared by the Knoevenagel condensation between 4-linked 2,2',6,6'-tetramethyl-4,4'-bipyridine (TMBP) unit and 2-linked Bpy-DA unit (Fig. 1d). The TMBP monomer was synthesized via a Yamamoto coupling reaction referred to previously reported literature (Supplementary Figs. 1 and 2)[33,34]. In a typical protocol, the experiment was carried out by sealing a mixed system of TMBP/Bpy-DA/benzoic anhydride/benzoic acid (1/2/2/0.2, molar ratio) in a glass tube and heating at 180 °C for 3 days to yield brown bulks (Supplementary Fig. 3). After the activation process to remove the unreacted monomers and impurities, a highly crystalline COF-205 with $sql$ topology was obtained as yellow powder (Fig. 1e and Supplementary Fig. 4). Subsequently, $RuCl_3$ was introduced into COF-205 framework and formed COF-205 with local Ru[bpy($H_2O$)$_x$Cl$_y$] coordinated structure (named as COF-205-RuO$_x$Cl$_y$, Fig. 1f). As displayed in Supplementary Fig. 5a, the uniformly distributed O and Cl elements in transmission electron microscopy (TEM) energy dispersive X-ray spectroscopy (EDS)-mapping images confirmed the initial coexistence of coordinated $H_2O$ and Cl. After 200 cyclic voltammetry (CV) cycles at 1.24-1.64 V (vs. RHE) under 0.5 M $H_2SO_4$, a significant loss of Cl element was observed (Supplementary Fig. 5b), along with the accumulation of oxygen element (the ratio of O/Cl is 1/0.07), confirming the conversion from the initial Ru–Cl bonds into Ru–O bonds (termed as COF-205-Ru, Fig. 1g)[19,35–38], which can be further demonstrated by the almost disappeared Cl $2p$ X-ray photoelectron spectroscopy (XPS) spectra after CV cycles (Supplementary Fig. 6).

### Structural investigation of COF-205 and COF-205-Ru

PXRD of COF-205 shown in Supplementary Fig. 7 indicates the disappearance of diffraction peaks for monomers, suggesting the network construction. Specifically, the dominant diffraction peaks located at 4.59°, 6.45°, 9.15° and 26.21° can be assigned to the (110), (200), (220) and (001) facets, matching well with the calculated indexes. In order to unravel the unit cell parameter, the profile of Pawley refinements was carried out on small-angle X-ray scattering (SAXS) for COF-205 (Fig. 2a). The refined result is in accordance with the collected data as evidenced by the satisfactory convergence residuals (*Cmmm* space group, orthorhombic lattice, $a = 27.346$ Å, $b = 27.083$ Å, $c = 3.276$ Å, and $\alpha = \beta = \gamma = 90°$, Rwp = 1.87%, and Rp = 1.04%). Notably, no obvious change of diffraction peaks is observed after the introduction of Ru into the COF-205 frameworks (Fig. 2b and Supplementary Fig. 8). Moreover, PXRD patterns of these COFs are consistent with the simulation of AA stacking model (Fig. 2c), which is obviously different from the staggered AB stacking model.

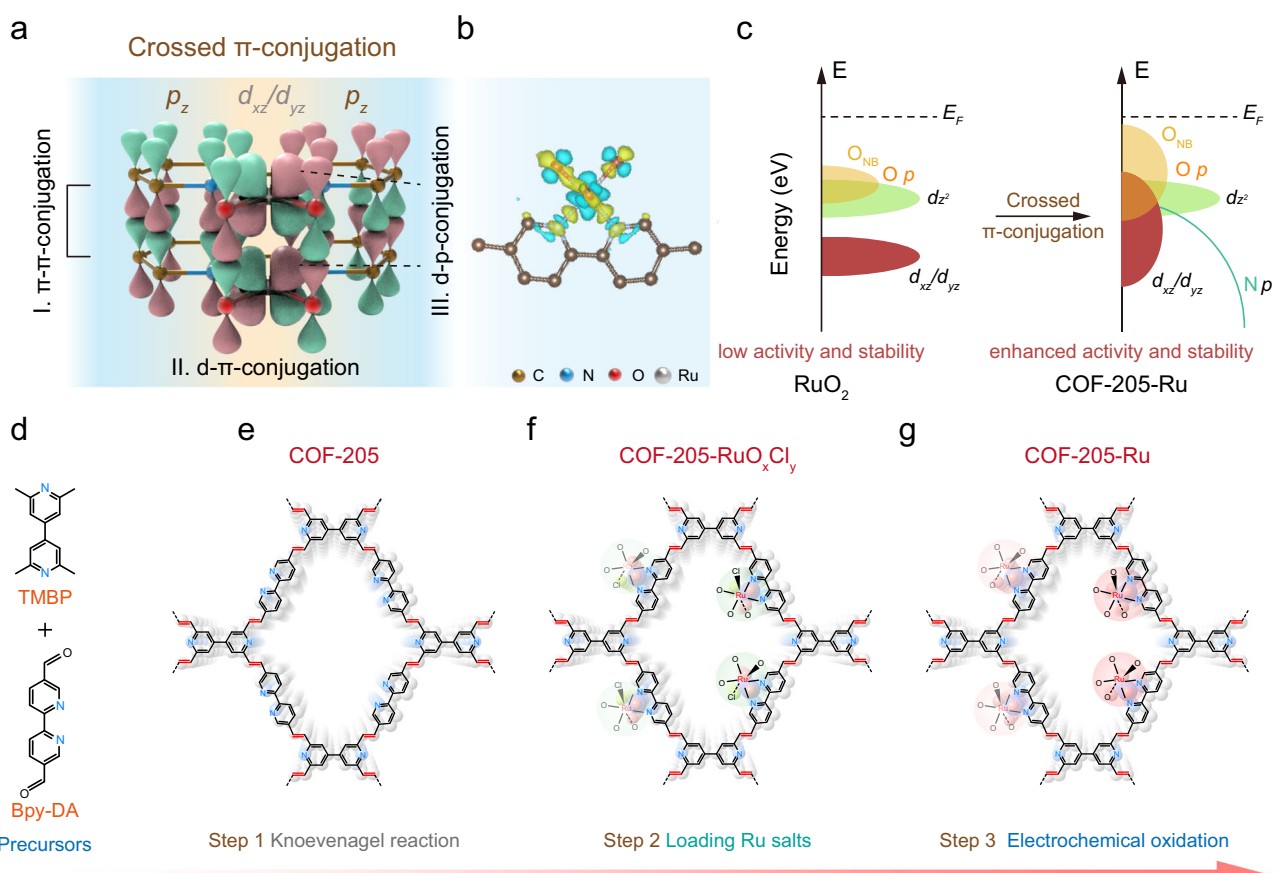

**Fig. 1 | Schematic illustration of the design and construction for COF-205-Ru and the crossed π-conjugation. a** The crossed π-conjugation system in COF-205-Ru. **b** Local charge density difference analysis in COF building block. **c** Schematic molecular orbital energy diagram for COF-205-Ru and RuO₂ toward the acidic OER. **d–g** The reaction monomers and synthesized process of the COF-205-Ru.

Spectroscopic studies were employed to verify the construction of $sp^2$-carbon linkage. Fourier transform infrared (FT-IR) spectra of COF-205, COF-205-Ru and corresponding monomers were investigated (Fig. 2d). The disappearance signals of C=O stretching vibration at 1700 cm⁻¹ for Bpy-DA and C−H vibration at 2916 cm⁻¹ for TMBP indicate the consumption of monomers. The newly formed peaks located at 961 and 1631 cm⁻¹ can be attributed to the *trans*-HC=CH and C=C stretch vibration, which clearly manifests a high polymerization degree of COF-205[39]. Moreover, as shown in Fig. 2e, the observation of the Raman peak located at about 1630 cm⁻¹, corresponding to the C=C stretching vibration, further confirms the successful formation of vinyl linkage. Besides, the ¹³C cross-polarization magic angle spinning solid-state NMR (¹³C CP-MAS NMR) signals of COF-205 and COF-205-Ru provide significant structure information to further support the construction of vinyl-linked COF (Fig. 2f)[33]. To our delight, both COF-205 and COF-205-Ru with robust $sp^2$-carbon linkage apparently provide excellent solvent stability and pH durability, even under the harshly acidic or basic environment for 2 weeks (Fig. 2g, and Supplementary Figs. 9, 10). Nitrogen adsorption–desorption measurements at 77 K were performed on COF-205 and COF-205-Ru to evaluate the permanent porosity. COF-205 displays a Type I isotherm with calculated pore size of around 1.6 nm, which is in good agreement with the pore size distribution of the simulated model (Fig. 2h). The Brunauer−Emmett−Teller (BET) surface area of COF-205 is calculated as 1380 m²/g (Supplementary Fig. 11), demonstrating the abundant accessible bipyridine sections to anchor Ru sites. For comparison, COF-205-Ru exhibits slightly decreased pore distribution (1.5 nm) and BET surface area (1200 m²/g) than pure COF-205, suggesting the successful anchoring of Ru (Fig. 2i and Supplementary Fig. 12). Moreover,

the TEM images of as-prepared COF-205 and COF-205-Ru show similar sheet-like morphology (Fig. 2j and Supplementary Fig. 13). Such 2D porous nanosheet-like structure generally offers easily accessible active sites, excellent conductivity and high BET surface area, which is favorable to the electrochemical behavior[40,41]. More importantly, as shown in Fig. 2k, the distinct lattice fringe spacing of 0.482 nm corresponding to the (220) plane of COF-205-Ru can be observed in high-resolution TEM (HRTEM) image, which in accordance with the PXRD results. In addition, the EDS maps of COF-205-Ru unveil the homogeneous and uniform distribution of C, N, Ru and O across the entire sample, further verifying the successful anchoring of Ru (Fig. 2l). This result was further confirmed by inductively coupled plasma-atomic emission spectroscopy (ICP−OES), in which the loading amount of atomically dispersed Ru is calculated as 6.8 wt% in COF-205-Ru.

## Activation of the coordinated oxygen

To further survey the chemical state, structural characteristics and electronic environment of the COF-205-Ru electrocatalyst, the XPS and X-ray absorption fine structure (XAFS) analysis were conducted. As shown in Fig. 3a, by comparison of the Ru K-edge X-ray absorption near edge structure (XANES) profiles of COF-205-Ru and RuO₂, the absorption edge shifts to the lower-energy position with the construction of crossed π-conjugation system in COF-205-Ru, suggesting the slightly decreased oxidation state of Ru, derived from the electron supply of π-conjugate system in COF-205 substrate (Fig. 3a). This change of Ru valence state was also evidenced by the XPS analysis (Fig. 3b). For deconvoluted Ru 3p XPS spectra, the peaks centered at binding energies of 464.5 and 486.9 eV are attributed to Ru³⁺, and the doublet with binding energies of 462.7 and 484.9 eV belong to Ru⁴⁺,

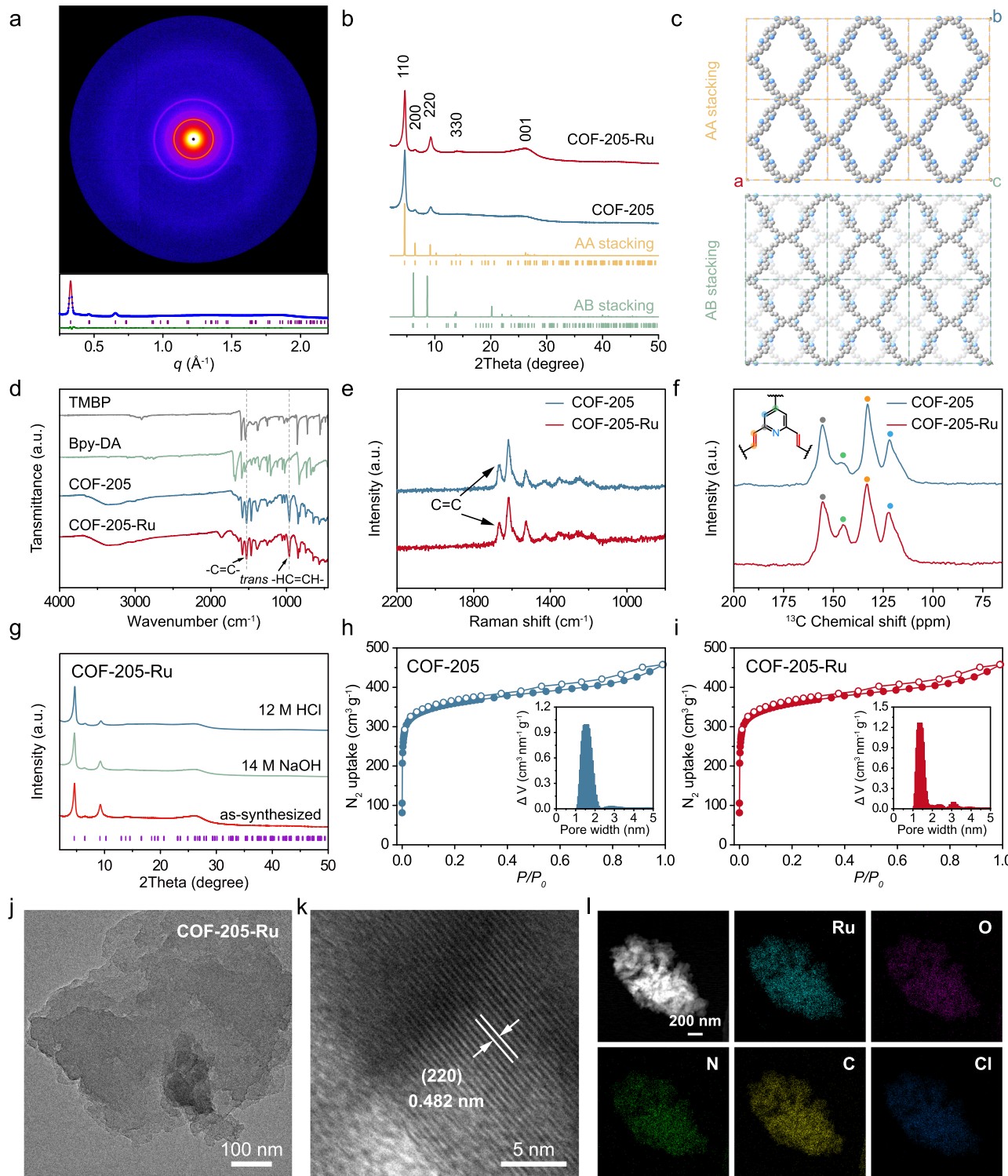

**Fig. 2 | The structural characterization of the as-prepared COF-205 and COF-205-Ru. a** 2D SAXS image and Pawley refinement of experimental SAXS data for the as-synthesized COF-205, where blue circles represent experimental data; red lines represent calculated data; green lines show the difference and purple bars show the Bragg position. **b** Experimental PXRD patterns of COF-205 and COF-205-Ru. **c** Simulated AA and AB stacking model for COF-205. **d** FT-IR spectra of COF-205, COF-205-Ru and corresponding monomers. **e** Raman spectra of COF-205 and COF-205-Ru. **f** $^{13}$C CP-MAS NMR signals of COF-205 and COF-205-Ru. **g** The pH durability evaluation for COF-205-Ru under 12 M HCl and 14 M NaOH. Nitrogen isotherm of COF-205 (**h**) and COF-205-Ru (**i**) at 77 K. Filled and open circles represent adsorption and desorption stages, respectively. Inset: corresponding pore size distribution. TEM (**j**), HRTEM (**k**) and TEM-EDS (**l**) images of COF-205-Ru. Scale bar: 100 nm for (**j**), 5 nm for (**k**) and 200 nm for (**l**), respectively.

respectively[42,43]. Moreover, the XPS peak intensity of Ru$^{4+}$ decreased from RuO$_2$ to COF-205-Ru, demonstrating the decreased Ru valence caused by electronic interaction between the COF framework and Ru species, this trend is consistent with the above XAS results.

Furthermore, as shown in Fig. 3c, the fitting results from XANES profiles reveal that Ru in COF-205-Ru presents an average valence state of +3.53. Compared with the O K-edge spectra of RuO$_2$, a decreased $t_{2g}/e_g$ intensity was observed in COF-205-Ru, which can be attributed to the

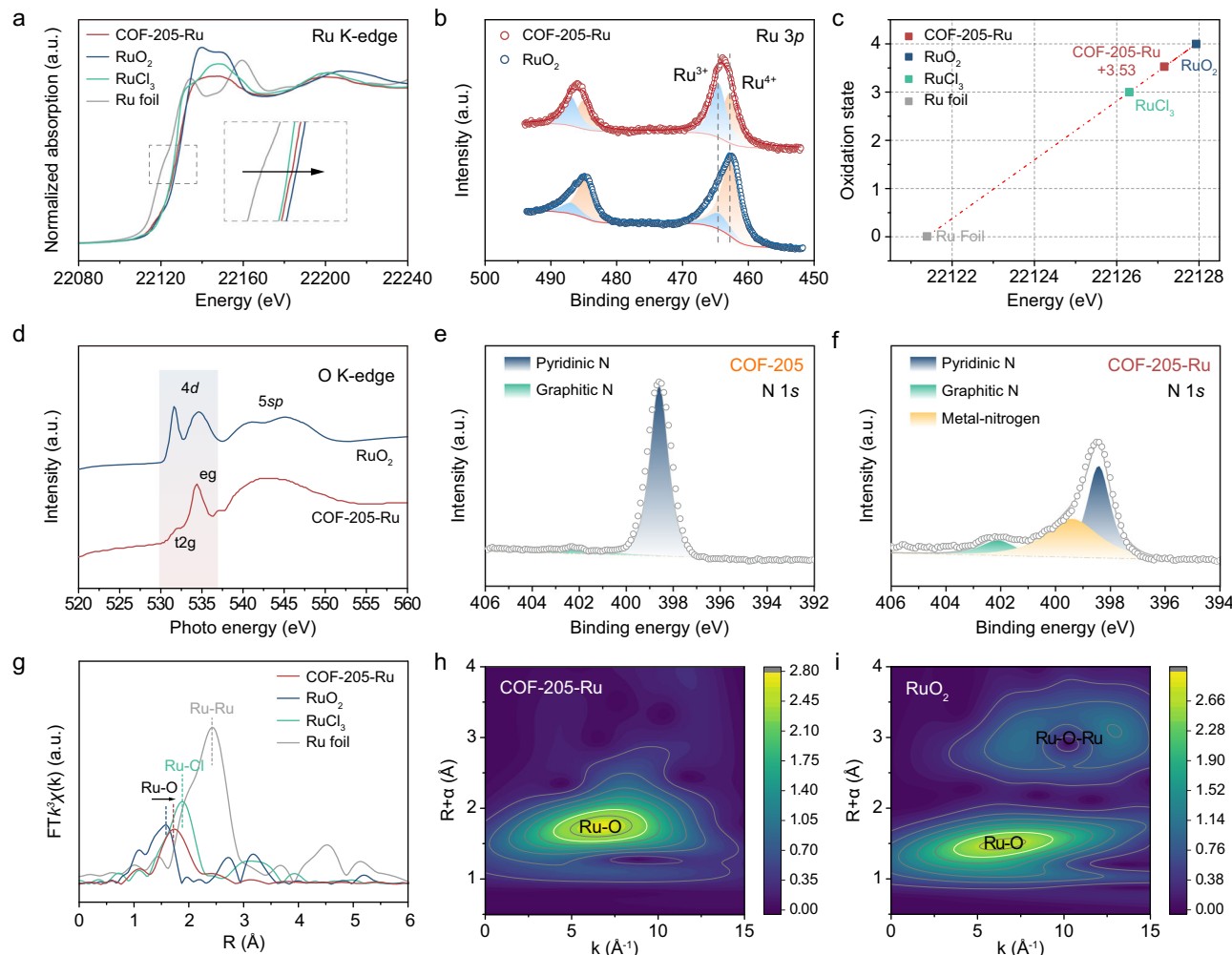

**Fig. 3 | Structural investigation of COF-205-Ru.** Ru K-edge XANES spectra (**a**), Ru 3*p* XPS spectra (**b**), calculated Ru oxidation state (**c**) for COF-205-Ru, RuO₂, RuCl₃ and Ru foil. **d** O K-edge spectra of COF-205-Ru and RuO₂. N 1*s* XPS spectra of COF-205 (**e**) and COF-205-Ru (**f**). **g** EXAFS of COF-205-Ru, RuO₂, RuCl₃ and Ru foil. Wavelet transform of Ru K-edge EXAFS data for COF-205-Ru (**h**) and RuO₂ (**i**).

reduced Ru oxidation state and increased O vacancies[44,45], underlying the important role of *d*–π interaction (Fig. 3d). The strong metal–substrate interaction could be also certificated by analyzing the N 1*s* spectra from XPS. For comparative purposes, the COF-205 displays an obvious peak at about 398.5 eV, corresponding to the pyridinic N signal (Fig. 3e)[46]. As expected, an additional peak located at about 399.5 eV corresponding to the metal–nitrogen signal (Fig. 3f) can be observed in COF-205-Ru[47], further confirming the strong Ru–N interaction. The calculated XPS peak area ratio of graphite N exhibits almost no change from COF-205 (10.2%) and COF-205-Ru (11.0%), demonstrating the well-maintained graphite N content after anchoring Ru. Moreover, we further supplement synchrotron radiation soft XANES measurement to investigate the atomic coordination structure of Ru and COF-205 framework. As displayed in Supplementary Fig. 14, a well-defined spectroscopic peak (N1, ~399 eV) observed in the N K-edge XANES spectra of COF-205 can be assigned to the pyridinic N 1*s* π* resonance[48,49]. The peak N₂ represents general transitions from N 1*s* core level to C–N σ* states[50,51]. It is worth noting that, compared to COF-205, an additional peak located at ~401 eV for COF-205-Ru could be observed, which can be assigned to Ru–N interaction, indicating the successful formation of Ru–N bonds. We speculate that the reduced Ru valence and robust Ru–N bond construction would weaken Ru–O interaction and facilitate the activation of coordinated oxygen to participate in the OER process. As shown in Fig. 3g, by analyzing the

extended X-ray absorption fine structure (EXAFS) data, no obvious peaks for Ru–Ru bonds and Ru–O–Ru bonds can be observed in COF-205-Ru, demonstrating the atomically dispersed Ru species without the formation of Ru–Cl bond. Notably, the COF-205-Ru exhibits an elongated Ru–O interatomic distance in comparison to RuO₂, which is caused by the decreased Ru valence and strong Ru–N interaction, implying the activation of oxygen. The local atomic structure around Ru sites was further investigated by fitting the EXAFS curves of COF-205-Ru in R-space (Supplementary Fig. 15). Specifically, the curve fittings results reveal an ~9% Ru–O bond length elongation in COF-205-Ru (2.16 Å) compared with RuO₂ (1.96 Å) (Supplementary Fig. 16 and Table 1). This geometrical distortion further leads to the rearrangement of Ru *d* orbital, which leads to enhanced hybridization degree between Ru-*d* orbital and O-*p* orbital[52,53], and increased Ru–O covalency (Supplementary Fig. 17)[53–60]. Besides, wavelet transform analysis of EXAFS spectra, which provides both R- and k-space information and discriminates the backscattering atoms, was employed to further distinguish the coordinated environment of Ru[61–63]. As shown in Supplementary Fig. 18, the horizontal axis shows the wave vector number *k*, which is the key to distinguishing different kinds of coordination atoms. The main intensity maximum at ~6.5 Å⁻¹ can be seen in the patterns of COF-205-Ru, which is almost identical to the position of RuO₂ and clearly different from that of RuCl₃, further proving the Ru–O coordinated structure. In addition, the elongation of Ru–O bond

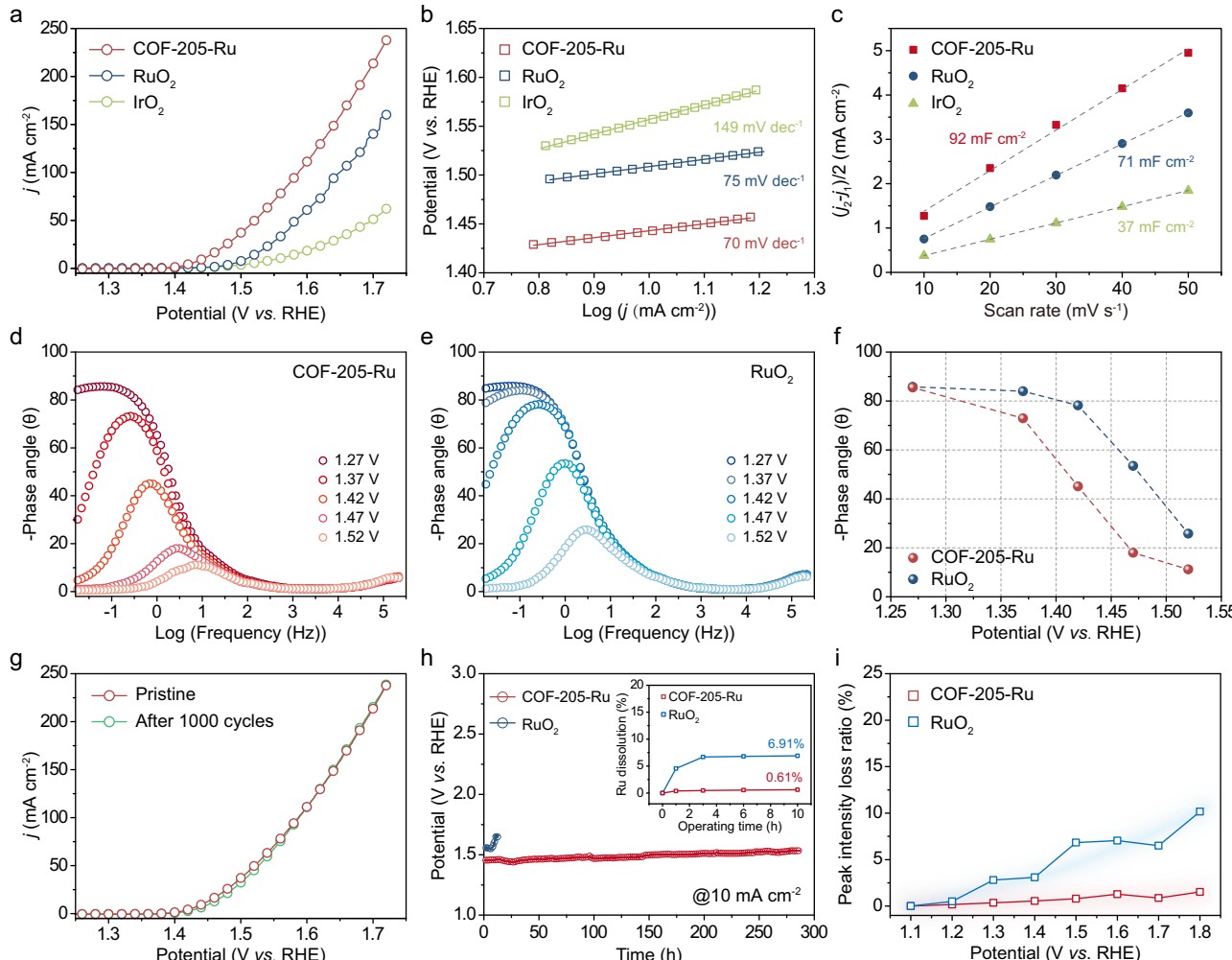

**Fig. 4 | Electrochemical OER evaluation for COF-205-Ru. a** Polarization curve of the as-prepared COF-205-Ru, with commercial $RuO_2$ and $IrO_2$ as comparison (without iR-compensation). **b** Corresponding Tafel plots. **c** Comparison of the $C_{dl}$ values of COF-205-Ru, $RuO_2$ and $IrO_2$. Bode phase plots of COF-205-Ru (**d**) and $RuO_2$ (**e**) at different applied potentials. **f** Summarized phase peak angles of COF-205-Ru and $RuO_2$ at 1.27–1.52 V. **g** Polarization curves of COF-205-Ru before and after 1000 CV cycles. **h** Chronopotentiometry measurements of COF-205-Ru and $RuO_2$ at the current density of 10 mA cm⁻² (inset: Ru dissolved mass fraction of COF-205-Ru and $RuO_2$ after operating times of 1, 3, 6 and 10 h). **i** Peak intensity loss ratio of main crystal plane for COF-205-Ru and commercial $RuO_2$ from in situ PXRD patterns.

length for COF-205-Ru than that of $RuO_2$, and the absence of Ru–Ru bond are observed, which are consistent with the EXAFS results (Fig. 3h, i). The anchoring of Ru sites in the COF-205 framework not only generates abundant atomically dispersed Ru sites but also optimizes the local electronic structure of Ru through the $d$–$\pi$ interaction, leading to the optimized Ru–O covalency and adjusted $d$-band configuration for enhanced OER kinetics.

**Electrocatalytic OER evaluation**

Encouraged by the unique structural investigation results, we further investigated the potential application for acidic OER. The electro-chemical OER performance of COF-205-Ru electrocatalyst was assessed by a standard three-electrode system under 0.5 M $H_2SO_4$ electrolyte, with commercial $RuO_2$ and $IrO_2$ for comparison (Supplementary Fig. 19). As shown in Fig. 4a, the COF-205-Ru displays a superior catalytic activity than that of commercial electrocatalysts, reflecting in the smallest required overpotential of 210 mV to deliver 10 mA cm⁻², superior to the benchmark $RuO_2$ (280 mV), $IrO_2$ (320 mV) and most of the documented Ru-based catalysts (Supplementary Table 2). Similarly, the COF-205-Ru also exhibits satisfactory OER performance on carbon cloth or Ti-mesh electrodes (Supplementary Figs. 20 and 21). Notably,

the activity contribution of pure COF-205 substrate is negligible (Supplementary Fig. 22) under the same condition, indicating the anchoring strategy significantly endowed Ru nodes with highly intrinsic activity. In addition, the COF-205-Ru shows the lowest Tafel slope (70 mV dec⁻¹) compared with commercial $RuO_2$ (75 mV dec⁻¹) and $IrO_2$ (149 mV dec⁻¹). The lower Tafel slope reveals enhanced OER kinetics for COF-205-Ru. It is worth noting that COF-205-Ru exhibits a high mass activity as 2659.3 A g⁻¹, which is 32-fold higher than commercial $RuO_2$ (82.3 A g⁻¹), and higher than most of the reported acidic OER electrocatalysts (Supplementary Table 3). Besides, the electrochemical surface area (ECSA) was assessed by calculating the double-layer capacitance ($C_{dl}$) (Supplementary Fig. 23). As shown in Fig. 4c, the $C_{dl}$ of COF-205-Ru is calculated to be 92 mF cm⁻², outperforming $RuO_2$ (71 mF cm⁻²) and $IrO_2$ (37 mF cm⁻²), demonstrates the abundant exposed active sites of COF-205-Ru favored by its high surface area, which is agreeing well with the above $N_2$ adsorption results (Fig. 2h). Moreover, benefitting from the crystalline 2D structure with fully conjugated framework, COF-205-Ru displays a lower charge transfer resistance ($R_{ct}$), superior to commercial $RuO_2$ and $IrO_2$, as evidenced by the electrochemical impedance spectroscopy (EIS) measurements (Supplementary Fig. 24). Furthermore, the in situ EIS measurement at different applied voltages was employed

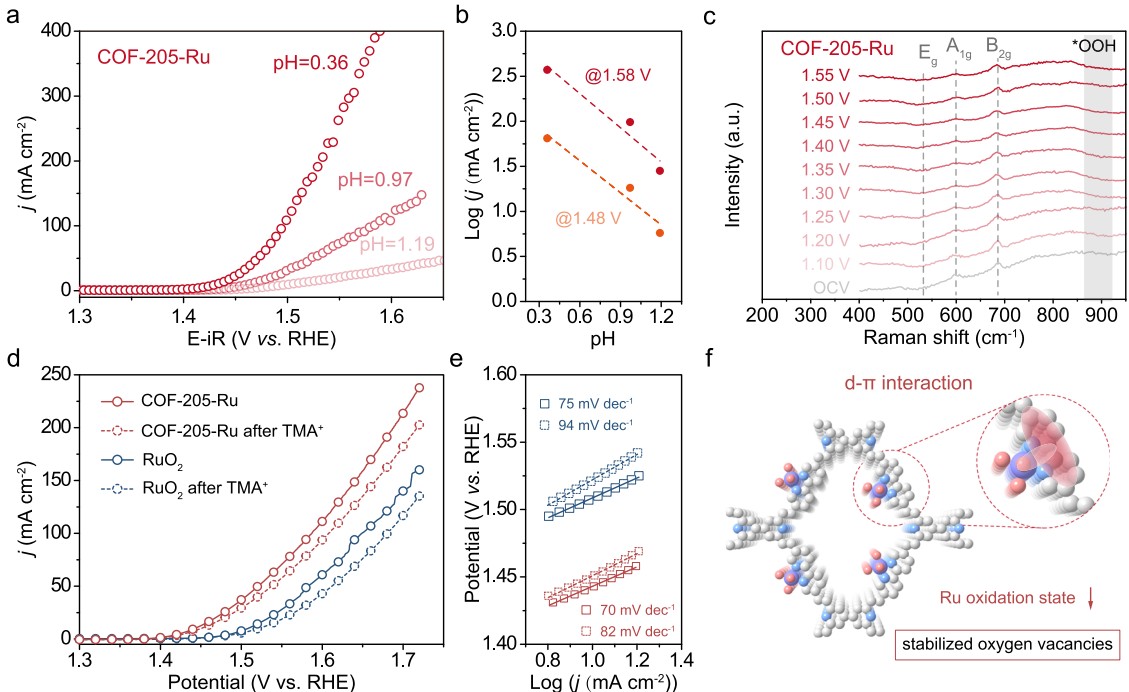

**Fig. 5 | Identification of mechanism and explanation of superior OER performance. a** pH dependence of the catalytic performance of COF-205-Ru with iR-compensation under $H_2SO_4$ (pH values: 0.36, 0.97 and 1.19, respectively). **b** Corresponding log($j$) at 1.60 and 1.70 V under different pH values. **c** In situ Raman spectra of COF-205-Ru electrode at different applied voltages in 0.5 M $H_2SO_4$ electrolyte. Polarization curves (**d**) and corresponding Tafel slopes (**e**) of COF-205-Ru and $RuO_2$ under 0.5 M $H_2SO_4$ electrolyte with or without adding TMA+ (without iR-compensation). **f** Schematic illustration of the $d$−π interaction to stabilize oxygen vacancy.

to investigate the electrocatalytic kinetics during the OER process. In the Bode phase plots, the phase angle peaks in high-frequency and low-frequency regions reflect in the intrinsic electron conduction of the electrocatalyst and the charge transfer at the electrolyte−catalyst interface, respectively[64]. As shown in Fig. 4d, e, both COF-205-Ru and $RuO_2$ exhibit higher phase peaks at low-frequency regions (≈0.01–10 Hz) than high-frequency regions (≈100–10,000 Hz), unveiling that the charge transfer is mainly limited by the electrolyte−catalyst interface resistance. With the increased applied potentials from 1.27 to 1.52 V, the phase angles at low frequency of COF-205-Ru show an accelerating trend of decline than that of $RuO_2$, indicating that the combination of Ru units with electroconductive COF-205 substrate can accelerate the electron transfer at electrolyte−catalyst interface and consequently lead to a superior OER kinetics (Fig. 4f). And the turnover frequency (TOF) values of COF-205-Ru and commercial $RuO_2$ were calculated based on the loading amounts of Ru to investigate the intrinsic activity (Supplementary Fig. 25). Specifically, the TOF value of COF-205-Ru is calculated to be $0.47\,s^{-1}$ at $\eta = 300\,mV$, much higher than that of $RuO_2$ ($0.014\,s^{-1}$). In addition, the faradaic efficiency for COF-205-Ru is nearly 100% (Supplementary Fig. 26), as determined by comparing the experimentally obtained oxygen volume with the theoretical expectation.

Combining the unique advantages of the fully conjugated structure, robust $sp^2$-carbon linkage and strong Ru−N interaction, it is expected that COF-205-Ru shows promising electrochemical stability. Considering the poor stability of commercial $RuO_2$ during acidic water oxidation, the durability of the as-prepared COF-205-Ru was evaluated in 0.5 M $H_2SO_4$. After 1000 cycles CV measurements, the obtained COF-205-Ru shows an unattenuated activity compared with the pristine state (Fig. 4g). Furthermore, COF-205-Ru exhibits nearly unaltered OER performance during the chronopotentiometric test over a period of 280 h at 10 mA $cm^{-2}$, which is better than the commercial $RuO_2$ (10 h) (Fig. 4h). Interestingly, the COF-205-Ru can operate stably for over 100 h under

0.5 M $H_2SO_4$ electrolyte even at current density of 50 mA $cm^{-2}$ (Supplementary Fig. 27). In addition, we quantified the Ru dissolution for COF-205-Ru and commercial $RuO_2$ during chronopotentiometry measurement (at 10 mA $cm^{-2}$) by ICP-MS analysis on electrolyte aliquots taken after 1, 3, 6 and 10 h (inset of Fig. 4h)[65]. As a result, the Ru dissolution in $RuO_2$ increases significantly (6.91% after 10 h), while that in COF-205-Ru is maintained well (0.61% increase after 10 h). Furthermore, in situ PXRD was performed to provide in-depth information and realistically assess the durability of COF-205-Ru at different applied potentials (Supplementary Fig. 28). To our delight, compared with commercial $RuO_2$, COF-205-Ru exhibits a much lower peak intensity loss ratio, further revealing its robust framework under OER operating potentials (Fig. 4i and Supplementary Fig. 29). In addition, the morphology, Ru content and valence state of COF-205-Ru after OER cycling test are maintained well, indicating the satisfactory durability (Supplementary Figs. 30–33).

We further investigated the reaction kinetics by the electrochemical and spectroscopic techniques to verify the pathway. We acquired the iR-corrected linear sweep voltammetry (LSV) curves of COF-205-Ru in $H_2SO_4$ electrolytes with different pH values (0.36, 0.97 and 1.19, respectively) at a scan rate of 10 mV $s^{-1}$ (Fig. 5a), with commercial $RuO_2$ (recognized LOM pathway) for comparison (Supplementary Fig. 34). Both the COF-205-Ru and $RuO_2$ exhibit obvious pH-dependent performances, indicating the non-concerted proton-electron transfer (NCPET) processes[58,66]. Besides, good linear relationships are observed between the corresponding log($j$) at different potentials with electrolyte pH values, underlying the coordinated oxygen-involved pathway (Fig. 5b and Supplementary Fig. 34b). In addition, to get insights into the OER mechanism, the electrochemical in situ Raman spectroscopy was employed to track the information of active species under OER conditions. Specifically, the Raman signals located at ~600 and 680 $cm^{-1}$ can be attributed to the $A_{1g}$ and $B_{2g}$ of $RuO_2$, respectively (Fig. 5c and Supplementary Fig. 35)[67]. In general, a distinct Raman band belonging to *OOH intermediates at ~900 $cm^{-1}$ can be detected if the

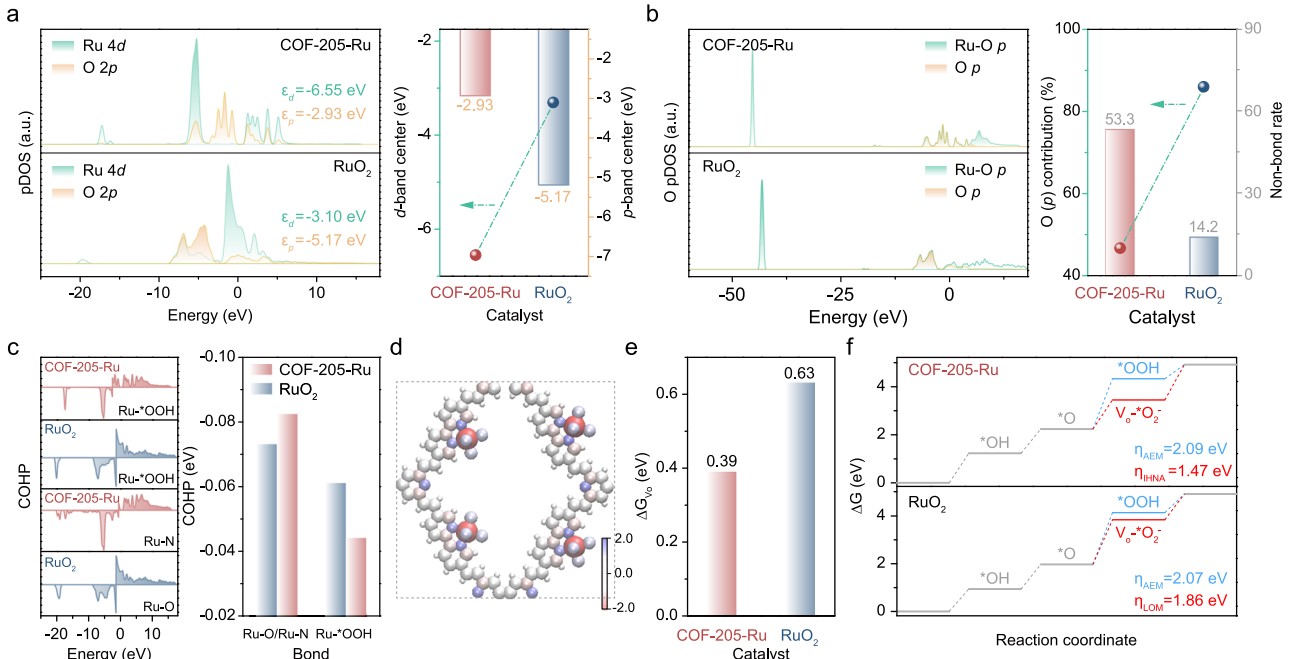

**Fig. 6 | DFT theoretical analysis for COF-205-Ru to explain the increased OER activity and stability. a** pDOS plots of Ru 4*d* and O 2*p* states for COF-205-Ru and RuO₂ (left); corresponding band centers (right). **b** O pDOS plots for COF-205-Ru and RuO₂. **c** COHP calculations of COF-205-Ru and RuO₂. **d** The calculated Bard charge analysis for the COF-205-Ru model. **e** Comparison of Gibbs free energy of $V_O$ for COF-205-Ru and RuO₂. **f** Gibbs free energy illustration for COF-205-Ru and RuO₂ during OER by AEM or IHNA pathways.

electrocatalyst performs the AEM pathway[68,69]. As shown in Fig. 5c, no obvious Raman band at ~900 cm⁻¹ can be observed for COF-205-Ru at OER potentials, which demonstrates the absence of *OOH species during the OER process, revealing the occurrence of the coordinated oxygen-involved pathway. Note that, for NCPET process, the proton transfer process may not be involved in the rate-limiting step, therefore, the kinetic isotope effects (KIE) investigation by H/D isotope-labeled experiment should be employed to reflect the proton transfer kinetic information and determine the reaction RDS[70–73]. As shown in Supplementary Fig. 36a, the use of D₂O decreases the activity of COF-205-Ru. The KIE values are calculated based on the reaction current density in the protonic vs. deuteric solution at the same overpotential (Supplementary Fig. 36b). The presence of KIEs (KIEs > 1.5) is considered as evidence that proton transfer is generally involved in the RDS[74–77], while COF-205-Ru exhibits KIE values less than secondary KIE (~1.5), indicating that proton transfer is not RDS. Moreover, referred to the previous reports, the tetramethylammonium cation (TMA⁺) is added into the electrolyte as a chemical probe to monitor the unique peroxo-like species (*O₂²⁻) from LOM path[78,79]. Specifically, the TMA⁺ could interact and overlay the negative *O₂²⁻ sites, thus resulting in the reduced kinetics of OER. As expected, after introducing TMA⁺, COF-205-Ru exhibits significantly decreased OER performance (Fig. 5d) and increased Tafel slope as 82 mV dec⁻¹ (Fig. 5e), which coincided with the RuO₂. It is worth noting that the precise structure of COF-205-Ru not only inhibits the dissolution of Ru through the strong Ru−N motifs, but also reduces the oxidation state of Ru by the *d*−π interaction, thereby leading to stabilized oxygen vacancies derived from coordinated oxygen departure, and promoted OER activity (Fig. 5f). Considering the possible reaction pathway on single Ru site, we proposed an intramolecular hydroxyl nucleophilic attack (IHNA) OER mechanism for the obtained COF-205-Ru catalyst (Supplementary Fig. 37)[18,20].

## DFT calculations
To get further insights into the nature of COF-205-Ru toward OER under acidic media, density functional theory (DFT) calculations were carried out (Supplementary Figs. 38 and 39). The density of state (DOS) for COF-205-Ru and RuO₂ were analyzed to investigate the rational regulation of Ru 4*d* and O 2*p* between Ru and COF-205. As shown in Fig. 6a, the much broader distribution of *d*-band and *p*-hole state in COF-205-Ru suggest the delocalization of electrons, which is attributed to the strong interaction of *d*−π conjugation, π−π conjugation and *d*−*p* conjugation[80–83]. The broad electron holes of *p*-band are much closer to the Fermi level ($E_F$), making coordinated oxygen atoms more electrophilic to facilitate the catalytic process[84–86]. Moreover, the O 2*p* band center moves from −5.17 eV (RuO₂) to −2.93 eV (COF-205-Ru), indicating the increased covalency of Ru−O bond in COF-205-Ru compared to RuO₂, thus lowering the energy barrier required for coordinated oxygen oxidation pathway[87]. Furthermore, the calculated O (*p*) contribution percentage decreased from 85.8% (RuO₂) to 46.7% (COF-205-Ru), also indicating the increased holes rate of O *p*-band, and strengthened Ru−O bond covalency in COF-205-Ru (Fig. 6b). Generally, the $O_{NB}$ states below $E_F$ could accommodate the electron from adsorbed oxygen species, leading to the formation of peroxo-like species. As shown in Fig. 6b, COF-205-Ru exhibits a much higher non-bond rate than RuO₂ counterpart, which is beneficial to promote the OER kinetics. Correspondingly, the bond length of Ru−O for optimized RuO₂ and COF-205-Ru model was investigated. As shown in Supplementary Fig. 40, COF-205-Ru exhibits an elongated Ru−O bond (2.060 Å) than that of RuO₂ (1.997 Å), in accordance with the XAS results, further indicating that coordinated oxygen is more conducive to participating during OER process. Additionally, the crystal orbital Hamilton population (COHP) of Ru−N bond in COF-205-Ru and that of Ru−O in RuO₂ were also investigated. The positive and negative COHP imply the anti-bonding states and the bonding states, respectively. As shown in Fig. 6c, the bonding states of Ru−N bond in COF-205-Ru are much higher than that of Ru−O in RuO₂, suggesting Ru−N is more stable than Ru−O, which is consistent with the calculated Bader charge analysis (Fig. 6d). Meanwhile, the bonding states of Ru−OOH bond in *OOH−COF-205-Ru are much lower under the Fermi level than Ru−OOH bond in *OOH−RuO₂, suggesting the adsorption ability of

*OOH in COF-205-Ru is much lower than that in $RuO_2$, thus verifying the favorable coordinated oxygen-involved pathway (Fig. 6c). Notably, compared with $RuO_2$ (0.63 eV), the COF-205-Ru displays the decreased Gibbs free energy of $V_O$ (0.39 eV), which is due to the optimizing of the local electronic structure and stabilization of the oxygen vacancies by the strong $d-\pi$ interaction between COF-205 substrate and Ru sites (Fig. 6e and Supplementary Fig. 41). Subsequently, the free energy profiles for COF-205-Ru and $RuO_2$ were calculated and analyzed to understand the enhanced OER kinetics (Fig. 6f and Supplementary Figs. 42–45). Both COF-205-Ru and $RuO_2$ show a lower RDS energy barrier for the IHNA pathway than the AEM pathway, agreeing well with experimental results. For LOM pathway, the RDS of $RuO_2$ is the formation of peroxo-like species (from *O to $V_O$-*$O_2^{2-}$), which is in coincidence with the previous literature. Benefitting from the activated Ru−O bond and stabilized oxygen vacancies by crossed π-conjugation systems, the energy barrier of RDS is significantly decreased for COF-205-Ru (1.47 V) compared to $RuO_2$ (1.86 V), which is responsible for the enhanced OER performance.

## Discussion

In summary, we have demonstrated that both the stability and activity of Ru-based catalysts can be simultaneously enhanced by atomically anchoring Ru sites on an acidic-stable π-conjugated COF with robust $sp^2$-carbon linkage. Benefitting from the unique porous structure of COF-205 with steady carbon linkage, and the atomically dispersed Ru units through strong Ru−N motifs, the obtained crossed π-conjugated COF-205-Ru exhibits high OER activity under acidic electrolyte, with the mass activity of 2659.3 A $g^{-1}$, 32-fold higher than the commercial $RuO_2$, and retain long-term durability of over 280 h. To the best of our knowledge, this is the first example of COF-based material for acidic OER application. Experimental results including XAS analysis, electrochemical investigation and in situ measurements reveal that the unique crossed conjugation in COF-205-Ru can not only activate the coordinated oxygen, stabilize the oxygen vacancies, and facilitate the coordinated oxygen-involved pathway, but also maintain the structural stability by robust frameworks and Ru−N motifs. DFT calculations suggest that the delocalization of electrons derived from strong interaction of $d-\pi$ conjugation, π−π conjugation and $d-p$ conjugation in COF-205-Ru, could effectively activate the Ru−O bond and stabilize the in situ formed oxygen vacancies, thereby leading to significantly decreased energy barrier of RDS for acidic OER process. From the perspectives of both fundamental science and practical applications, this work provides an effective strategy to simultaneously solve the activity and stability bottleneck of Ru-based electrocatalysts for acidic OER.

## Methods
### Chemicals
4-Bromo-2,6-lutidine, $NiBr_2(PPh_3)_2$, Zn powder, $Et_4NI$, and 2,2′-bipyridyl-5,5′-dialdehyde (BPy-DA) were purchased from Shanghai Aladdin Bio-Chem Technology Co., Ltd. Benzoic acid and benzoic anhydride were purchased from Adamas Reagent, Ltd. Anhydrous ruthenium (III) chloride ($RuCl_3$) was purchased from Changcheng Chemical Co., Ltd. Nafion® 117 solution (~5% in a mixture of lower aliphatic alcohols and water) was obtained from Sigma-Aldrich Co., Ltd. Tetramethylammonium chloride (TMACl) and sulfuric acid ($H_2SO_4$) were purchased from Sinopharm Chemical Reagent Co., Ltd. All the common solvents including, petroleum ether, $n$-hexane, dichloromethane, ethyl acetate, acetone, acetonitrile, tetrahydrofuran, methanol, ethanol, etc. were also purchased from Sinopharm Chemical Reagent Co., Ltd. and used as received without further purification. The ultrapure water (18.25 MΩ cm$^{-1}$) prepared from an up water purification system (Ulupure) was used throughout the whole experiment.

### Synthesis of 2,2′,6,6′-tetramethyl-4,4′-bipyridine (TMBP)
TMBP monomer was synthesized according to the literature procedure[33,34]. Typical procedure as shown in Supplementary Fig. 1: 4-bromo-2,6-lutidine (4.60 g, 25.0 mmol), $NiBr_2(PPh_3)_2$ (5.58 g, 7.5 mmol), Zn powder (2.50 g, 38.0 mmol) and $Et_4NI$ (6.42 g, 25.0 mmol) were mixed and refluxed in super dry-THF solution (15 mL) for 24 h under Ar atmosphere. After cooling to room temperature, the mixture was filtrated, and then the solvent was evaporated to obtain a crude residue. The residue was treated with 10% aqueous solution of ethylenediamine and then extracted $CH_2Cl_2$. An aqueous solution of HCl (1 M) was added to the organic layer. After separation, the aqueous layer was treated with an aqueous solution of NaOH (1 M). When the extraction ($CH_2Cl_2$) was dried over $MgSO_4$ and evaporated, the residue was further purified by column chromatography (EA:PE = 1:1) to give the model compound as a white solid.

### Synthesis of COF-205
The COF-205 was obtained by a classic Knoevenagel condensation reaction. In a typical synthesis, a Pyrex tube was filled with TMBP (84.92 mg, 0.4 mmol), BPy-DA (169.8 mg, 0.8 mmol), benzoic acid (9.8 mg, 0.08 mmol) and benzoic anhydride (181.0 mg 0.8 mmol) sequentially. The mixture was degassed through three freeze–pump–thaw cycles. After gradual warm up to room temperature, the tube was placed in an oven and heated at 180 °C for 3 days. The bulk monolith was then soaked in a mixture of 1 M aq. NaOH and methanol (v/v = 1/1) for 24 h. The solid was cracked into powder by a hammer and ground by a mortar. The obtained powder was subsequently fluxed with acetone/methanol (v/v = 1/1) in a Soxhlet extractor for 12 h. After drying under a dynamic vacuum at room temperature overnight, the yellow powder was collected.

### Synthesis of COF-205-Ru
The COF-205-Ru was obtained by a simple anchoring process. In a typical synthesis, COF-205 (100 mg), $RuCl_3$ (30 mg), 15 mL deionized water and 5 mL THF were evenly distributed in a 20 mL Pyrex vial by ultrasonic treatment. The mixture was heated in a 90 °C oven for 6 h. After cooling down to room temperature, the obtained powder was sequentially washed with deionized water and ethanol. After drying under a dynamic vacuum at room temperature overnight, the black powder was collected. The collected powder was dispersed to a fluorine-doped tin oxide-coated glass electrode and underwent 200 CV cycles at a range of 1.24-1.64 V under 0.5 M $H_2SO_4$ electrolyte to prepare COF-205-Ru.

### Materials characterizations
The PXRD measurements were performed on a Rigaku SmartLab 9 KW diffractometer (45 kV, 200 mA) with Cu Kα (λ = 1.5406 Å) target under room temperature. PXRD data were collected with a scan speed of 6° min$^{-1}$. The simulated PXRD patterns were calculated from the CIF files of COF-205 and COF-205-Ru using Materials Studio 8.0. The SAXS data were acquired on a Rigaku 3.5 m NANOPIX (45 kV, 56 mA) with Cu Kα (λ = 1.5406 Å) target under room temperature. The camera length was calibrated using silver behenate as a reference before measurements. In situ electrochemical PXRD experiments were performed on Rigaku SmartLab 9KW diffractometer. PXRD data were collected with a scan speed of 8° min$^{-1}$ at 2$\theta$ range of 2–30° and 10–40° for COF-205-Ru and commercial $RuO_2$, respectively. Carbon cloth was chosen as a conductive carrier to support the catalysts. In situ Raman measurements were carried out utilizing a home-made electrochemical cell on a HORIBA Raman microscope with grating parameter: 600 (750 nm), and laser intensity: 25%. A roughened Au disk electrode with loaded catalyst' ink was employed as a working electrode. The Raman spectra were collected over the potential range of 1.10–1.55 V at 50 mV intervals. Solution nuclear magnetic resonance measurement was carried out on a Bruker AVANCE III HD 400 MHz spectrometer with

tetramethyl silane as the internal reference using deuterated DMSO as solvent. The FT-IR spectra were collected on a Thermo Nicolet iS10 IR spectroscope with a scanning wavelength range from 4000 to 400 cm$^{-1}$. Samples were tableted with mixing KBr as a background. The $^{13}$C cross-polarization/magic angle spinning solid-state nuclear magnetic resonance experiments were carried out using a Bruker AVANCE NEO 400 MHz NMR spectrometer operating at 100.63 MHz for $^{13}$C with a double resonance 4 mm MAS NMR probe. The sample spinning rate was set as 10 kHz, with a cross-polarization duration of 3 ms. $^{13}$C MAS chemical shifts adamantine (39 ppm). The XPS measurements were conducted using a Thermo Fisher ESCALAB 250Xi, employing a monochromatic Al Kα X-ray source. All the binding energies were calibrated with the characteristic C 1$s$ peak at 284.8 eV. And the energy scales were calibrated with the Fermi level of the XPS instrument (4.57 eV vs. absolute vacuum value). The TEM images were obtained from a JEOL JEM-2100Plus TEM, which operated at an acceleration voltage of 200 kV. ICP-MS measurements were performed on a Perkin-nElmer NexION 5000G to quantify the Ru dissolution of catalysts during chronopotentiometry measurement. There were 100 mL of 0.5 M H$_2$SO$_4$ in the initial electrochemical cell, and 8 mL of the electrolyte was taken at 1, 3, 6 and 10 h for stability tests to determine the dissolution of Ru, without adding additional electrolyte to avoid diluting the solution. XAFS spectra were collected from BL14W1 beamline of Shanghai Synchrotron Radiation Facility. The collected XAFS data were processed in Athena (version 0.9.26) for data calibrations (background, pre-edge, post-edge and energy shift). And the Fourier-transformed data were further fitted in Artemis (version 0.9.26) to monitor the coordinated information for samples.

## Preparation of working electrodes

For OER measurements, 3 mg COF-205-Ru, 1 mg acetylene carbon and 0.6 mL 0.1wt% Nafion solution were ultrasonicated over 30 min to obtain homogeneous ink. The glassy carbon electrode (GCE, 5 mm in diameter, 0.19625 cm$^2$ in disk area) was polished with 0.05 μm γ-Al$_2$O$_3$ powders, washed with absolute ethanol by sonication and dried under air to obtain a neat surface. Subsequently, 18 μL ink was dropped on the surface of the GCE (loading: ~0.45 mg cm$^{-2}$$_{geo}$) and dried under air for further electrochemical tests.

## Electrochemical characterizations

All electrochemical tests were performed on a CHI 760E electrochemical workstation with a three-electrode system under O$_2$-saturated 0.5 M H$_2$SO$_4$ at room temperature. The GCE loaded with electrocatalysts served as the working electrode, the graphite rod served as the counter electrode and the Hg/HgO electrode served as a reference electrode. The GCE loaded with catalysts was employed as the working electrode, the saturated calomel electrode (SCE, Hg/Hg$_2$Cl$_2$) was employed as a reference electrode and the graphite rod was employed as a counter electrode. In this work, all of the offered potentials were converted into the reversible hydrogen electrode (RHE) potential according to the following formula: E (vs. RHE) = E (vs. SCE) + 0.059*pH + 0.242. Polarization curves were collected using a rotation disk electrode with a rotation speed of 1600 rpm, with the potential scan range from 1.0 to 1.5 V (vs. SCE) under 0.5 M H$_2$SO$_4$. The chronopotentiometric stability tests were evaluated in 0.5 M H$_2$SO$_4$ using a standard three-electrode system with carbon cloth or Ti-mesh electrode. CV tests were carried out in the non-Faradaic region, employing a range of scan rates of 10, 20, 30, 40 and 50 mV s$^{-1}$. The evaluation of ECSA was based on the $C_{dl}$ value, which was obtained by plotting the $\Delta j/2$ ($\Delta j = j^a − j^c$, where $j^a$ and $j^c$ are anodic current and cathodic current at the middle potential) against the scan rate, where the slope is $C_{dl}$ value. EIS measurements were performed with the AC impedance spectra at a frequency range of 100 kHz to 10 mHz with an amplitude of 5 mV. The $R_{ct}$ values were processed in Z View, the equivalent fitted circuit diagram and

impedance values were indicated in Supplementary Fig. 24. The LSV curves of pH dependence measurements were directly collected with 80% iR-compensation from the workstation's built-in function (CHI 760E). To study the mechanism of the water oxidation reaction, the TMA$^+$ was added into the electrolyte as a chemical probe to monitor the unique peroxo-like species (*O$_2$$^{2-}$). Specifically, (CH$_3$)$_4$NCl (101 mg, 1 mmol) in 2 mL of deionized water was evenly added into the 0.5 M H$_2$SO$_4$ electrolyte (10 mM TMA$^+$). Subsequently, the polarization curves of COF-205-Ru and RuO$_2$ were collected and analyzed. The TOF values, reflecting the activity of a single catalytic site, were calculated by Eq. (1):

$$TOF = (j \times A)/(4 \times F \times n) \tag{1}$$

where $j$ (mA cm$^{-2}$) is the current density at an applied potential; $A$ is the GCE geometric area (0.19625 cm$^2$), $F$ is the Faraday's constant (86,485 C mol$^{-1}$) and $n$ is the molar number of active sites. In this work, the number $n$ (mol) was calculated with Eq. (2):

$$n = (m \times N_A)/M_w \tag{2}$$

where $m$ is the Ru loading mass, $N_A$ and $M_w$ are the Avogadro's constant and the molecular weight, respectively.

## DFT calculations

**RuO$_2$**. DFT computations were executed utilizing the generalized gradient approximation combined with the Perdew–Burke–Ernzerhof (PBE) functional for estimating exchange-correlation energy. To handle core-valence interactions effectively, ultrasoft pseudopotentials were employed. These calculations were carried out using the CASTEP module within the Materials Studio software package, developed by Accelrys Inc. The cutoff energy and the self-consistent field (SCF) tolerance were set as 440 eV and 1 × 10$^{-6}$ eV, respectively. The RuO$_2$ surface was selected with a 4*4*3 unit cell and the vacuum width was chosen as 15 Å. In the simulation, the lowest layer was held stationary, whereas the upper two layers along with the adsorbates were permitted to undergo positional relaxation. A grid of 7*7*1 Monkhorst–Pack mesh k-points was employed for all surface calculations. The energy differences computed across all planes were controlled to remain within a tight tolerance of 0.01 eV. The atomic coordinates of the optimized computational models for RuO$_2$ can be seen in Supplementary Data 1.

**COF-205-Ru**. The DFT computations were executed utilizing the generalized gradient approximation combined with the PBE functional for estimating exchange-correlation energy. The cutoff energy was set as 500 eV, and the geometries were fully relaxed until the SCF tolerance was 1 × 10$^{-6}$ eV. The Brillouin zone integration was performed with 2*2*1 Monkhorst–Pack k-point sampling for a primitive cell that Ru complexes coordinate to bpy building blocks. The atomic coordinates of the optimized computational models for COF-205-Ru can be seen in Supplementary Data 2.

In this work, the electrochemical processes were investigated from the established computational hydrogen electrode (CHE) model. The premise for validating the CHE model is that electrochemical processes occur in solution typically encounter minimal kinetic obstacles that are easily overcome at standard room temperature conditions. Therefore, the reaction kinetics can be dictated by the free energy difference of each reaction step, and the RDS is the step with the maximum free energy difference. The theoretical overpotential ($\eta$) can be calculated by Eq. (3):

$$\eta_{theory} = \frac{max\{\Delta G_1, \Delta G_2, \Delta G_3, \Delta G_4\}}{e} - 1.23V \tag{3}$$

## Data availability

Source data are provided with this paper.

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

## Acknowledgements

This work was financially supported by the National Natural Science Foundation of China (22272121, 21972107), and the Natural Science Foundation of Hubei Province (2020CFA095). We thank the Core Facility of Wuhan University for the measurement of XPS and ICP-MS. We also thank the Core Research Facilities of the College of Chemistry and Molecular Sciences for the measurement of TEM. We sincerely thank the great help from Prof. L. Zhuang, Prof. H. Deng and co-workers at Wuhan University for helpful measurements and discussion. We thank the Beamlines MCD-A and MCD-B (Soochow Beamline for Energy Materials) at NSRL for O and N K-edge measurements and analysis. DFT calculations in this paper have been done on the supercomputing system in the Supercomputing Center of Wuhan University.

## Author contributions

W.L. conceived and supervised the project. H.J. synthesized the electrocatalysts. H.J., N.Y., L.W. and J.Z. completed the ligand synthesis and performed the catalytic tests and characterization. N.Y. and Y.J. performed the density functional theory calculations. W.L., H.J. and N.Y. wrote the manuscript. All the authors discussed the results and assisted during the manuscript preparation.

## Competing interests

The authors declare no competing interest.
