## [Peer Review File · Nature Communications]

Stabilizing atomic Ru species in conjugated sp² carbon-linked covalent organic framework for acidic water oxidationREVIEWER COMMENTS

Reviewer #1 (Remarks to the Author):

This is a very bizarre paper. The most bizarre here is that authors claim forming RuO_x without any proof they made such and with synthetic protocol unlikely to result in such.

Their synthetic protocol decorates Ru into bpy fragments. This process is well established: RuCl₃ readily reacts with nitrogen containing ligands. Authors should cite Ezhov et al. 2023 ChemSusChem and ACS Catalysis 2020 for similar systems. It is very hard to understand how RuO_x came into the picture. As far as I can track it, the material labeled "COF-205-Ru" did not undergo any additional treatments (per Materials and Methods) after the synthesis.

"As shown in Fig. 3g, by analyzing the extended X-ray absorption fine structure (EXAFS) data, no obvious peaks for Ru-Ru bonds and Ru-O-Ru bonds can be observed in COF-205-Ru, demonstrating the atomically dispersed Ru oxides." – No, it does not demonstrate "the atomically dispersed Ru oxides" (there is no such thing as "atomically dispersed oxide" – this is a chemically absurd wording. There can, at best, be a nano-particles of RuO₂). In this case EXAFS demonstrates the molecular Ru complex with bpy ligand, some water and some Cl ligands. Authors should show a more comprehensive EXAFS fits. For instance, to show if any Cl ligands to Ru are present and if any Ru=O bonds are present. I did not find a Table with EXAFS fits.

Glassy carbon electrode used in this study corrodes under water oxidation conditions. Authors should use something else.

Discussion of AEM pathway vs LOM pathway is irrelevant as no solid-state phase of RuO_x was demonstrated.

Authors explicitly misrepresent their data. DFT was done on "RuO₂ surface was selected with repeated in 4x4x3 unit cell and a vacuum width of 15 Å was used." BUT a Figure 6 shows Ru attached to the COF structure. This is very confusing and must not appear in any publication. Minor comment: Figure 2I – nothing is visible for Ru distribution (black square).

Basically, authors need to get back to basic chemistry and fully re-write this paper if they wish to publish with any qualified reviewer.

Reviewer #2 (Remarks to the Author):

In this study, the authors reported a novel electrocatalyst by anchoring Ru oxides into an acidic stable COF for OER in acidic water. Through the introduction of Ru-N motifs and unique crossed π -conjugation structure, the oxygen vacancies are stabilized and the lattice oxygen are activating, which suppress the dissolution of Ru during the LOM pathway. Overall, more measurements should be performed and some discussion should be reorganized to further improve the quality of the manuscript. Some detailed concerns are listed as below:

1. The manuscript mentioned that the crossed π -conjugation structure suppresses the dissolution of Ru, but the experiment data is not enough to support this argument. The ICP-OES results of after stability tested RuO₂ should be provided to illustrate it (i.e. the dissolved amount of Ru for COF-205-Ru and RuO₂ during stability test).
2. While the electrocatalyst have a long-term durability of 125 h, I wonder did the electronic structure of COF-205-Ru change during the stability test. XAFS or XPS of after stability tested COF-205-Ru is suggested to provide to monitor the valence state change during OER. Moreover, characterizations to monitor the change of ligand are also necessary.
3. The Ru 3p XPS spectra shows that the binding energy of Ru 3p_{3/2} is positively shifted compared to that of RuO₂ and Ru foil. How could this reveal that the valence state of Ru in COF-205-Ru is between 0 and +4? Related detailed discussion should be provided.
4. How to explain that the N 2p XPS spectra of COF-205 and COF-205-Ru show an increase of Graphitic N after anchoring Ru?

5. According to the EXAFS that authors use to support their claim seems to be contradictory to the reference. In the reference (J. Am. Chem. Soc. 143, 6482-6490 (2021)), Ir-O bond length have a shrinkage, indicating the increased covalency Ir-O bond.
6. The order of Fig.2g-2i do not correspond to the manuscript(L162-169).

Reviewer #3 (Remarks to the Author):

In this manuscript, the authors have successfully developed a robust conjugated COF framework to encapsulate Ru sites for acidic OER electrocatalysis, which demonstrated simultaneously improved stability and activity compared to the commercial RuO₂. Interestingly, the COF-205-Ru can not only facilitate the LOM pathway by activating the lattice oxygen and stabilizing the oxygen vacancies, but also maintain the structural stability by robust frameworks and Ru-N motifs. Different with conventional elemental-doping engineering, this work provides a new direction to develop efficient OER catalysts. Overall speaking, the work is well organized and written, the results would be enlightening. As a result, it could be published in this journal after addressing some questions below.

1. Please provide more details besides XPS to further prove the Ru-N interaction.
2. To get more insight into the reaction kinetics mechanism, the authors should supplement KIE investigation by H/D isotope-labelled experiment.
3. I am interested in conjugated sp² carbon-linked COF-205-Ru to achieve efficient and stable acidic OER. In this view, stability test at large current densities (such as 50 mA cm⁻²) or long-term V-T test (more than 200 h) should be added to further support the stable feature of the catalysts.
4. For pH-dependence experiments, the different solution resistance (R_s) for these H₂SO₄ electrolytes have to considered.
5. After the stability test, the authors use in situ XRD, TEM and ICP to illustrate the well-maintained phase structure, morphology and metal content, I think it is necessary to supplement characterization to verify whether the metal valence of COF-205-Ru changed after OER test.
6. The authors are required to provide turnover frequency (TOF) values of COF-205-Ru and other samples to evaluate catalytic performance.
7. As shown in Figure 4c, the authors have compared the activity with other recently reported acidic OER catalysts. However, the displayed overpotential of COF-205-Ru is about 320 mV, please correct it.
8. In situ XRD should use attenuation ratio of characteristic diffraction peak rather than stack lines. As a comparison, the in situ XRD data of commercial RuO₂ deserve to be supplemented.

Reviewer #1:

This is a very bizarre paper.

Response: We are sincerely grateful for your valuable time and effort spent in reviewing our manuscript. These comments are constructive and valuable to improve our manuscript and future research direction. We have revised our manuscript through cautious supplementary analysis and discussion, and the specific responses are as follows. We hope that our response addresses your concerns.

Question 1. The most bizarre here is that authors claim forming RuO_x without any proof they made such and with synthetic protocol unlikely to result in such. Their synthetic protocol decorates Ru into bpy fragments. This process is well established: RuCl₃ readily reacts with nitrogen containing ligands. Authors should cite Ezhov et al. 2023 ChemSusChem and ACS Catalysis 2020 for similar systems. It is very hard to understand how RuO_x came into the picture. As far as I can track it, the material labeled “COF-205-Ru” did not undergo any additional treatments (per Materials and Methods) after the synthesis.

Response: We are sincerely grateful for your constructive comment.

1). We are grateful for the reviewer pointing out these nice works reported the important Ru(bpy) catalysts towards photocatalytic water oxidation reaction (WOR) under acidic electrolyte (pH = 1). Importantly, Ezhov and co-workers integrated [Ru(bpy)(dcbpy)(H₂O)₂]²⁺ building block into the chelating framework of MOF (like UiO-67) to avoid the deactivating dimerization and realize efficient WOR kinetics with water nucleophilic attack mechanism (*ACS Catal.* **2020**, *10*, 5299–5308). Moreover, they further promoted the scope of the combination of [Ru(bpy)(dcbpy)(H₂O)₂]²⁺ catalytic unit and Fe-based MOF (Fe-MIL-126) to investigate the synergistic behavior. They found that Fe₃O nodes emerge as photosensitizers able to drive prolonged O₂ evolution in acid (*ChemSusChem* **2023**, *16*, e202202124). We highly value the recommended reference, as it undoubtedly holds significant value to design the molecular Ru electrocatalysts and understand the interaction between Ru sites and

framework substrate. Accordingly, we have cited these excellent reports in **Ref. 18-19** in the revised manuscript.

2). As you mentioned, decorating Ru into bpy fragments of COF-205 framework is well established: RuCl₃ readily reacts with nitrogen containing ligands. Indeed, 2,2'-bipyridine (2,2'-Bpy) unit is commonly employed in complexation due to its robust redox stability and ease functionalization (*Chem. Rev.* 2000, **100**, 3553–3590). As we know, the dangling Ru-Cl bonds are easily oxidized, the coordinated Cl can be replaced by H₂O only in the presence of air and water (*ChemSusChem* **2023**, *16*, e202202124; *J. Am. Chem. Soc.* **2017**, *139*, 17747–17750). In addition, electrocatalysts have been found to undergo a structural reconstruction during the OER process (*Chem. Rev.* **2021**, *121*, 13174–13212; *Adv. Mater.* **2021**, *33*, 2007344). For instance, the etching of lattice anion from cobalt oxychloride (Co₂(OH)₃Cl) and the formation of Co-O bond were observed during OER process (*Adv. Mater.* **2019**, *31*, 1805127). Note that, in our manuscript, we explained the follow-up procedures “the cyclic voltammetry (CV) oxidation was employed to provide an oxidation potential to convert the possible Ru-Cl bonds into Ru-O bonds” (as shown in **Figure R1**). Therefore, it is reasonable to understand the disappearance of Ru-Cl coordination and the formation of Ru-O coordination after CV oxidation process. To further clarify this process, as shown in **Figure R2** (*Figs. 1d-1g* in Revised Manuscript), we supplemented the synthetic details of the COF-205-Ru catalyst and added the corresponding discussion.

3). According to your comments, we have performed some additional experiments and discussion in order to track the structure of COF-205-Ru clearly. Firstly, the RuCl₃ was anchored into COF-205 framework and formed local Ru[bpy(H₂O)_xCl_y] coordinated structure (named as COF-205-RuO_xCl_y). As displayed in **Figure R3a** (*Supplementary Fig. 5a* in Revised Supplementary information), the uniformly distributed O and Cl elements in TEM EDS-mapping images confirmed the initial coexistence of coordinated H₂O and Cl, demonstrating the structure of COF-205-RuO_xCl_y. However, after 200 CV cycles at 1.24 V ~ 1.64 V under 0.5 M H₂SO₄, a significant loss of Cl element was observed (**Figure R3b**), along with the accumulation

of oxygen element (the calculated ratio of O/Cl is 1/0.07), confirming the conversion from the initial Ru-Cl bonds into Ru-O bonds (termed as COF-205-Ru), which can be further demonstrated by the almost disappeared Cl 2p XPS spectra after CV cycles (Figure R4, Supplementary Fig. 6 in Revised Supplementary information).

4). From the perspective of basic chemistry, we considered that the statement about “Ru oxides” is not accurate. In fact, the local catalytic unit is formed by Ru site and coordinated O and/or H₂O, thus, we have revised the relevant unrigorous expressions and highlighted in the revised manuscript. Moreover, We have corrected the TITLE of the manuscript to **“Stabilizing atomic Ru species in conjugated sp² carbon-linked covalent organic framework for acidic water oxidation”** accordingly.

Figure R1. The synthetic procedure of COF-205-Ru electrocatalyst.

Figure R2. The reaction monomers and synthesized process of the COF-205-Ru.

Figure R3. TEM-EDS images of COF-205-Ru (a) before and (b) after CV cycles.

Figure R4. Cl 2p XPS spectra of COF-205-Ru before and after CV cycles.

Accordingly, **Figure R2** has been integrated into Revised Manuscript as Figure 1. **Figures R3** and **R4** have been added in Supplementary Information as **Supplementary Figs. 5** and **6**, respectively. The relevant experimental details and discussion have been supplemented in the Revised Manuscript as follow:

“The collected powder was dispersed to fluorine doped tin oxide (FTO) coated glass electrode and underwent 200 CV cycles at a range of 1.24 V ~ 1.64 V under 0.5 M H₂SO₄ electrolyte to prepare COF-205-Ru.” (the highlighted text in the **Methods** section, Page 18)

*“from a structural viewpoint, precise constructing a strong metal-support interaction is a promising strategy to stabilize Ru catalytic units (ACS Catal. **2020**, *10*, 5299–5308; ChemSusChem **2023**, *16*, e202202124; Nat. Catal. **2022**, *5*, 414–429).”* (the highlighted text in the **Introduction** section, Page 2)

“Subsequently, RuCl₃ was introduced into COF-205 framework and formed COF-205 with local Ru[bpy(H₂O)_xCl_y] coordinated structure (named as COF-205-RuO_xCl_y,

*Fig. 1f). As displayed in Supplementary Fig. 5a, the uniformly distributed O and Cl elements in TEM EDS-mapping images confirmed the initial coexistence of coordinated H₂O and Cl. After 200 CV cycles at 1.24 V ~ 1.64 V (vs. RHE) under 0.5 M H₂SO₄, a significant loss of Cl element was observed (Supplementary Fig. 5b), along with the accumulation of oxygen element (the ratio of O/Cl is 1/0.07), confirming the conversion from the initial Ru-Cl bonds into Ru-O bonds (termed as COF-205-Ru, Fig. 1g)^{19,35-38}, which can be further demonstrated by the almost disappeared Cl 2p XPS spectra after CV cycles (Supplementary Fig. 6).” (the highlighted text in the **Principle of crossed π -conjugation for COF-205-Ru** section, Page 5)*

Question 2. “As shown in Fig. 3g, by analyzing the extended X-ray absorption fine structure (EXAFS) data, no obvious peaks for Ru-Ru bonds and Ru-O-Ru bonds can be observed in COF-205-Ru, demonstrating the atomically dispersed Ru oxides.” – No, it does not demonstrate “the atomically dispersed Ru oxides” (there is no such thing as “atomically dispersed oxide” – this is a chemically absurd wording. There can, at best, be a nano-particles of RuO₂). In this case EXAFS demonstrates the molecular Ru complex with bpy ligand, some water and some Cl ligands. Authors should show a more comprehensive EXAFS fits. For instance, to show if any Cl ligands to Ru are present and if any Ru=O bonds are present. I did not find a Table with EXAFS fits.

Response: Thanks very much for your constructive question and it is useful to improve the quality of this work. Indeed, as discussed above, from the perspective of basic chemistry, we considered that the statement about “Ru oxides” is not accurate. In fact, the atomically dispersed catalytic unit is formed by Ru site and coordinated O and/or H₂O (which can be confirmed by the unobserved Ru-Ru and Ru-O-Ru bonds from EXAFS data), thus, we have revised the relevant unrigorous expressions and highlighted in the revised manuscript. In addition, the Ru-Cl bonds in COF-205-RuO_xCl_y are easily oxidized, the coordinated Cl can be replaced with O or H₂O, which was confirmed by TEM-EDS mapping and XPS results.

Moreover, it is necessary to further investigate the local coordinated structure of COF-205-Ru by XAS analysis. As you suggested, in order to identify if any Cl ligands

to Ru are present and if any Ru=O bonds are present, we collected the XAS spectra of RuCl₃ sample as a comparison and reanalyzed the XAS results. As shown in **Figure R5** (*Fig. 3* in Revised Manuscript), the absorption edge of COF-205-Ru is between RuCl₃ and RuO₂, and the fitting results from XANES profiles reveal that Ru in COF-205-Ru presents average valence state of +3.53. Moreover, as shown in **Figure R5c**, by analyzing the EXAFS data, no obvious peaks for Ru-Ru and Ru-O-Ru can be observed in COF-205-Ru, demonstrating the atomically dispersed Ru nodes and no Ru-Cl coordinated structure can be found. Notably, the COF-205-Ru exhibits an elongated Ru-O interatomic distance in comparison to RuO₂, which is caused by the decreased Ru valence and Ru-N interaction, implying the activation of Ru-O bonds. The local atomic structure around Ru sites was further investigated by fitting the EXAFS curves of COF-205-Ru in R-space (**Figure R6**, *Supplementary Fig. 15* in Revised Supplementary information). Specifically, the curve fittings results reveal an approximately 9% Ru-O bond length elongation in COF-205-Ru (2.19 Å) compared with RuO₂ (1.99 Å) (Table R1). More importantly, to further distinguish the coordinated environment of Ru, wavelet transform (WT) analysis of EXAFS spectra, which provides both R- and k-space information and discriminates the backscattering atoms, was carried out (*Angew. Chem. Int. Ed.* **2022**, *61*, e202212329; *Nat. Energy* **2020**, *5*, 881–890; *Nat. Commun.* **2023**, *14*, 6849; *Adv. Mater.* **2024**, *36*, 2310699). As shown in **Figure R7** (*Supplementary Fig. 18* in Revised Supplementary information), the horizontal axis shows the wave vector number k, which is the key to distinguish different kinds of coordination atoms. Atoms with small atomic numbers are weak in scattering photoelectrons, and their strongest oscillations will occur in the low k portion, while atoms with large atomic numbers are the opposite, and their strongest oscillations will occur in the high k portion. The main intensity maximum at ~6.5 Å⁻¹ can be seen in the patterns of COF-205-Ru, which is almost identical to the position of RuO₂ and clearly different from that of RuCl₃, further proving the Ru-O coordinated structure.

Figure R5. (a) XANES and (b) EXAFS spectra of COF-205-Ru with Ru foil, RuO₂ and RuCl₃ as comparison.

Figure R6 Ru K-edge EXAFS (point) and curvefit (line) for (a) Ru foil, (b) RuCl₃ (c) RuO₂ and (d) COF-205-Ru shown in R-space.

Figure R7. Wavelet transformation of COF-205-Ru, RuO₂ and RuCl₃ samples.

Table R1. Structural parameters extracted from the Ru K-edge EXAFS fitting. ($S_0^2=0.70$).

Sample	Path	CN	R (Å)	σ^2 (10^{-3} Å ²)	ΔE_0 (eV)	R factor
Ru foil	Ru-Ru	12*	2.67 ± 0.02	2.6	4.4	0.01
RuO ₂	Ru-O	5.6 ± 0.8	1.99 ± 0.03	3.4	-0.8	0.02
RuCl ₃	Ru-Cl	5.7 ± 0.9	2.34 ± 0.02	4.3	-2.96	0.02
COF-205-Ru	Ru-N	1.7 ± 0.3	2.05 ± 0.03	1.0	9.3	0.02
	Ru-O	3.4 ± 0.6	2.19 ± 0.02	1.7	9.3	0.02

For the EXAFS fitting, S_0^2 is the amplitude reduction factor; CN is the coordination number; R is interatomic distance (the bond length between Ru central atoms and surrounding Ru atoms or coordination oxygen atoms); σ^2 is Debye-Waller factor (a measure of thermal and static disorder in absorber-scatterer distances); ΔE_0 is edge-energy shift (the difference between the zero kinetic energy value of the sample and that of the theoretical model). R factor is used to value the goodness of the

fitting. S_0^2 was fixed according to the experimental EXAFS fit of Ru foil by fixing CN as the known crystallographic value.

Accordingly, **Figure R5** has been integrated into Revised Manuscript as **Fig. 3**. **Figures R6** and **R7** have been added in Supplementary Information as *Supplementary Figs. 15 and 18* in Revised Supplementary information, respectively. The relevant experimental details and discussion have been supplemented in the Revised Manuscript as follow:

“demonstrating the atomically dispersed Ru species without the formation of Ru-Cl bond.

*Besides, wavelet transform (WT) analysis of EXAFS spectra, which provides both R- and k-space information and discriminates the backscattering atoms, was employed to further distinguish the coordinated environment of Ru (Nat. Energy **2020**, 5, 881–890; Nat. Commun. **2023**, 14, 6849). As shown in Supplementary Fig. 18, the horizontal axis shows the wave vector number k , which is the key to distinguish different kinds of coordination atoms. The main intensity maximum at $\sim 6.5 \text{ \AA}^{-1}$ can be seen in the patterns of COF-205-Ru, which is almost identical to the position of RuO_2 and clearly different from that of RuCl_3 , further proving the Ru-O coordinated structure.”* (the highlighted text in the *Activation of the coordinated oxygen* section, Page 10)

Question 3. Glassy carbon electrode used in this study corrodes under water oxidation conditions. Authors should use something else.

Response: Thank you for your specialized question. As you mentioned, the glassy carbon electrode is not the best support for long-term stability measurement during OER process. Seitz et al. found that glassy carbon is not electrochemically inert under OER conditions on the timescale of common stability tests, which can cause electrodes to exhibit performance losses that do not reflect the intrinsic stability of the actual catalyst material being investigated (*ACS Appl. Energy Mater.* **2022**, 5, 12206). In fact, we considered this issue and all the chronopotentiometric stability were evaluated in 0.5 M H_2SO_4 using a standard three-electrode system with carbon cloth (CC) electrode

in this work (**Figures R8-R9**, *Fig. 4h and Supplementary Fig. 24* in Revised version). We apologize for missing this detail in the previous submission and have supplemented corresponding experimental details for stability testing in the revised Method section.

As for LSV measurement, following OER standard measurements (*Nat. Catal.* **2019**, *2*, 304–313; *J. Am. Chem. Soc.* **2021**, *143*, 6482–6490. *Nat. Commun.* **2019**, *10*, 162; *Nat. Commun.* **2020**, *11*, 5368; *Nat. Mater.* **2023**, *22*, 100–108; *Nat. Commun.* **2023**, *14*, 1412), we first evaluated OER activities of the COF-205-Ru catalyst and control samples on a RDE set-up. In a typical test, electrochemical measurements were evaluated in an O₂-saturated 0.5 M H₂SO₄ electrolyte with a three-electrode configuration by a CHI 760E workstation. The cell contained a catalyst-modified glassy carbon electrode (GCE, 5 mm in diameter, disk geometric area, 0.196 cm²), a saturated calomel electrode (SCE), and a graphite rod as the working electrode, the reference electrode, and the counter electrode, respectively. An RDE assembly with the prepared glassy carbon electrode were used as the working electrode, at a rotation rate of 1,600 r.p.m.

In addition, we further investigated the electrochemical performance of COF-205-Ru and commercial RuO₂ on CC electrode. As shown in **Figure R10** (*Supplementary Fig. 19* in Revised Supplementary information), LSV curves showed that the as-synthesized COF-205-Ru presented better OER activity than RuO₂, the overpotential for COF-205-Ru to reach 10 mA cm⁻² was 216 mV, lower than that of RuO₂ (300 mV), indicating its superior OER kinetics.

Figure R8. Chronopotentiometry measurements of COF-205-Ru on CC electrode at current density of 10 mA cm⁻².

Figure R9. Chronopotentiometry measurements of COF-205-Ru on CC electrode at current density of 50 mA cm^{-2} .

Figure R10. OER performance of COF-205-Ru and commercial RuO_2 on carbon cloth electrode under $0.5 \text{ M H}_2\text{SO}_4$ electrolyte.

Accordingly, **Figure R8** has been integrated into Revised Manuscript as **Fig. 4**. **Figures R9** and **R10** have been added in Supplementary Information as **Supplementary Figs. 24** and **19**, respectively. The relevant experimental details and discussion have been supplemented in the Revised Manuscript as follow:

“Similarly, the COF-205-Ru also exhibits superior OER performance than RuO_2 counterpart in carbon cloth electrode (Supplementary Fig. 19).” (the highlighted text in the **Electrocatalytic OER evaluation** section, Page 11)

“All the chronopotentiometric stability tests were evaluated in $0.5 \text{ M H}_2\text{SO}_4$ using a standard three-electrode system with carbon cloth electrode.” (the highlighted text in the **Methods** section, Page 19)

“Interestingly, the COF-205-Ru can operate stably for over 100 h under 0.5 M

H₂SO₄ electrolyte even at current density of 50 mA cm⁻², representing excellent acidic OER stability (Supplementary Fig. 24).” (the highlighted text in the *Electrocatalytic OER evaluation* section, Page 13)

Question 4. Discussion of AEM pathway vs LOM pathway is irrelevant as no solid-state phase of RuO_x was demonstrated.

Response: Thank you for your constructive question. YES, we agree with the reviewer about this part, and it is very useful to improve the quality of this work.

1). As discussed above, the structure of COF-205-Ru has been confirmed as the Ru sites with coordinated O and/or H₂O embedded in COF-205 framework by robust Ru-N interaction. The Ru unit is not RuO_x compound, but an atomically dispersed Ru molecular catalyst, similar structures have been widely reported in previous reports (*Chem. Mater.* 2016, 28, 4375–4379; *Nat. Catal.* **2022**, 5, 414–429). Once again, we sincerely apologize for this imprecise statement about “RuO_x”, and we have revised the corresponding expression in the revised manuscript.

2). In fact, efficient electrocatalytic/photocatalytic water oxidation with MOF/COF anchored single-atom metal sites is a relatively mature strategy (*Chem. Mater.* 2016, 28, 4375–4379; *ACS Catal.* **2020**, 10, 5299–5308; *Nat. Catal.* **2022**, 5, 414–429). For example, Ezhov et al anchored Ru complex into MOF frameworks toward water oxidation and confirmed the formation of the highly active Ru^V=O key intermediate, following a water nucleophilic attack (WNA) mechanism (**Figure R11a**). In addition, Sun and co-workers reported a O-O coupling process in an Aza-fused, π-conjugated, microporous polymer (Aza-CMP) coordinated single cobalt sites (Aza-CMP-Co) during alkaline OER process (**Figure R11b**), which proves the feasibility of the intramolecular hydroxyl nucleophilic attack (IHNA) mechanism catalyzed by single atomic sites. These results encourage us to further survey the mechanism source of the efficient water oxidation achieved by COF-205-Ru. **Figure R12** (*Supplementary Fig. 32* in Revised Supplementary information) shows the proposed IHNA process on Ru single site of the obtained COF-205-Ru catalyst.

3). In this work, we evaluated the reaction pathway of COF-205-Ru by pH-dependent experiment, *in situ* Raman spectroscopy, TMA⁺ molecular probe experiment and so on, with commercial RuO₂ (recognized LOM pathway) for comparison (*J. Am. Chem. Soc.* **2021**, *143*, 6482-6490; *Energy Environ. Sci.* **2022**, *15*, 1119-1130; *Nat. Commun.* **2023**, *14*, 1412; *Angew. Chem. Int. Ed.* **2024**, *63*, e202316029; *ACS Catal.* **2020**, *10*, 3650-3657; *ACS Catal.* **2022**, *13*, 256-266). Both the COF-205-Ru and RuO₂ exhibit obvious pH-dependent performances, indicating the non-concerted proton-electron transfer (NCPET) processes (*Nat. Chem.* **2017**, *9*, 457-465; *Catal. Today* **2016**, *262*, 2-10). In addition, no obvious Raman band at ~900 cm⁻¹ can be observed for COF-205-Ru and RuO₂ at OER potentials, which demonstrates the absence of *OOH species during the OER process (*Nat. Commun.* **2023**, *14*, 7115; *ACS Catal.* **2017**, *7*, 7873–7889), revealing the occurring of coordinated oxygen involved pathway. Finally, after introducing TMA⁺, COF-205-Ru exhibits significantly decreased OER performance and increased Tafel slope, which coincided with the RuO₂, underlying the coordinated oxygen involved pathway.

Figure R11. (a) Reaction pathway for water oxidation of $\text{cis}[\text{Ru}(\text{bpy})(5,5'\text{-dcbpy})(\text{H}_2\text{O})_2]^{2+}$ according to monomolecular mechanism^[ACS Catal. **2020**, *10*, 5299–5308]; (b) Proposed OER mechanisms of WNA (similar with AEM, blue) and IHNA (similar with LOM, red) pathways^[Nat. Catal. **2022**, *5*, 414–429].

Figure R12. Schematic illustration of the reaction pathway for COF-205-Ru catalyst.

Accordingly, **Figure R12** has been added into Revised Supplementary Information as **Supplementary Fig. 35**. The relevant discussion has been supplemented in the Revised Manuscript as follow:

*“Sun and co-workers reported a O-O coupling process in an Aza-fused, π -conjugated, microporous polymer coordinated single cobalt sites (Aza-CMP-Co) during alkaline OER process, which proves the feasibility of the intramolecular O-O coupling mechanism by single atomic sites in COF structure (Nat. Catal. **2022**, 5, 414–429). Unfortunately, when the electrolyte turns to acidic condition, traditional COFs with conventional imine linkages usually suffer from the collapse of structure (Nat. Rev. Chem. **2022**, 6, 881-898; Angew. Chem., Int. Ed. **2021**, 60, 4705-4711; Angew. Chem., Int. Ed. **2023**, 62, e202306135).”* (the highlighted text in the **Introduction** section, Page 3)

*“Considering the possible reaction pathway on single Ru site, we proposed an intramolecular hydroxyl nucleophilic attack (IHNA) OER mechanism for the obtained COF-205-Ru catalyst (Supplementary Fig. 35) (Chem. Mater. 2016, 28, 4375–4379; Nat. Catal. **2022**, 5, 414–429; ACS Catal. **2020**, 10, 5299–5308).”* (the highlighted text in the **Electrocatalytic OER evaluation** section, Page 15)

Question 5. Authors explicitly misrepresent their data. DFT was done on “RuO₂ surface was selected with repeated in 4x4x3 unit cell and a vacuum width of 15 Å was used.”

BUT a Figure 6 shows Ru attached to the COF structure. This is very confusing and must not appear in any publication.

Response: Thanks for your careful consideration and professional question. In the DFT calculations' part (Supplementary information), we only describe the modeling steps of RuO₂, missing the modeling steps of COF-205-Ru, and we sincerely apologize for this. According to your constructive suggestion, we have supplemented the modeling steps of COF-205-Ru as follow.

COF-205-Ru: Wave functions were expanded using generalized gradient approximation and Perdew–Burke–Ernzerhof (PBE) functional with kinetic energy cutoff of 500 eV and the geometries were fully relaxed until the self-consistent field (SCF) tolerance was 1×10^{-6} eV. The Brillouin zone integration was performed using $2 \times 2 \times 1$ Monkhorst-Pack k-point sampling for a primitive cell that Ru complex coordinates to two N ligands of bpy unit.

Specifically, as discussed above, according to previous reports about molecular metallic complexes in MOF/COF structure, metal sites are decorated into bpy fragments (*ACS Catal.* **2020**, *10*, 5299–5308; *ChemSusChem* **2023**, *16*, e202202124; *J. Am. Chem. Soc.* **2017**, *139*, 17747–17750; *Coord. Chem. Rev.* **2021**, *445*, 214050). In addition, reports typically employed unit cell as the DFT models to calculate metal functionalized COF complex towards OER electrocatalysis (*Angew. Chem. Int. Ed.* **2022**, *61*, e202213522; *J. Am. Chem. Soc.* **2022**, *144*, 38, 17661–17670; *ACS Catal.* **2022**, *12*, 9101–9113). Considering the coordination environment of Ru sites, the initial coordinated Cl ligands in Ru[bpy(H₂O)_xCl_y] are easily oxidized to be replaced with O or H₂O. As Ezhov et al reported, stirring of [Ru(bpy)(5,5' - dcbpy)]Cl₂-UIO-67 with water for 12 h leads to a substitution of Cl ligands with water molecules (*ACS Catal.* **2017**, *7*, 7873–7889). Therefore, considering the applied OER potential and combining the experimental results, as shown in **Figure R13** (**Supplementary Fig. 36** in the Revised Supplementary information), we performed the unit cell of COF-205-Ru without coordinated Cl as the DFT model to complete the

calculations. In addition, we have supplemented the geometric configurations and the intermediates adsorption on COF-205-Ru and RuO₂ with different potential reaction pathway (AEM and IHNA) in the revised manuscript (Figures R14-R17, Supplementary Figs. 40-43 in the Revised Supplementary information).

Figure R13. The geometric configuration on the calculated COF-205-Ru model.

Figure R14. The geometric configuration of COF-205-Ru with the adopted adsorption sites of (a) *OH₂, (b) OH*, (c) *O, and (d) *OOH on Ru site following AEM pathway. The dark cyan, light grey, blue, red, and white balls represent Ru, C, N, O, and H atoms, respectively.

Figure R15. The geometric configuration of COF-205-Ru with the adopted adsorption sites of (a) *OH₂, (b) OH*, (c) *O, and (d) *OO_{lat} on Ru site following IHNA pathway. The dark cyan, light grey, blue, red, and white balls represent Ru, C, N, O, and H atoms, respectively.

Figure R16. The geometric configuration of RuO_2 with the adopted adsorption sites of (a) $^*\text{OH}_2$, (b) OH^* , (c) $^*\text{O}$, and (d) $^*\text{OOH}$ on Ru site following AEM pathway. The dark cyan, red, and white balls represent Ru, O, and H atoms, respectively.

Figure R17. The geometric configuration of RuO_2 with the adopted adsorption sites of (a) $^*\text{OH}_2$, (b) OH^* , (c) $^*\text{O}$, and (d) $^*\text{OO}_{\text{int}}$ on Ru site following LOM pathway. The dark cyan, red, and white balls represent Ru, O, and H atoms, respectively.

Accordingly, **Figures R14-R17** has been added into Revised Supplementary Information as **Supplementary Fig. 40-43**. And the relevant details have been supplemented in the Revised Supplementary information as follow:

*“COF-205-Ru: Wave functions were expanded using generalized gradient approximation and Perdew–Burke–Ernzerhof (PBE) functional with kinetic energy cutoff of 500 eV and the geometries were fully relaxed until the self-consistent field (SCF) tolerance was 1×10^{-6} eV. The Brillouin zone integration was performed using $2 \times 2 \times 1$ Monkhorst-Pack k-point sampling for a primitive cell that Ru complex coordinates to two N ligands of bpy unit.” (the highlighted text in the **DFT calculations** section, Page 19)*

Question 6. Minor comment: Figure 21 – nothing is visible for Ru distribution (black square).

Response: Thanks for your careful inspection and we sincerely apologize for the confusion caused by using BLUE color as the Ru signal in TEM-EDS mapping. The enlarged TEM-EDS mapping image is displayed in **Figure R18**, COF-205-Ru unveils the homogeneous and uniform distribution of C, N and Ru across the entire sample, which verifies the successful anchoring of Ru. To avoid confusing readers, we replaced previous Figure 21 with the recollected TEM-EDS mappings (**Figure R19**, *Fig. 21* in Revised Manuscript), the O and Cl signals were added to illustrate the loss of Cl element.

Figure R18. TEM-EDS images of COF-205-Ru. Scale bar: 200 nm.

Figure R19. TEM-EDS images of COF-205-Ru. Scale bar: 200 nm.

Accordingly, **Figure R16** has been added into Revised Manuscript as **Fig. 2**.

Question 7. Basically, authors need to get back to basic chemistry and fully re-write this paper if they wish to publish with any qualified reviewer.

Response: Firstly, we once again sincerely thank you for your professional review and valuable time in reviewing our manuscript. Based on your constructive suggestions, we

have carefully checked and fully revised our manuscript. The main revised contents include that i) the structure evolution of Ru-Cl coordination; ii) the structure investigation of COF-205-Ru; iii) the electrochemical activity and stability in carbon electrode; iv) kinetics mechanism discussion and v) some experimental or computational detail. After careful revision based on your valuable comments, we believe that the quality of the manuscript has been effectively improved and we hope that our response addresses your concerns.

Reviewer #2:

In this study, the authors reported a novel electrocatalyst by anchoring Ru oxides into an acidic stable COF for OER in acidic water. Through the introduction of Ru-N motifs and unique crossed π -conjugation structure, the oxygen vacancies are stabilized and the lattice oxygen are activating, which suppress the dissolution of Ru during the LOM pathway. Overall, more measurements should be performed and some discussion should be reorganized to further improve the quality of the manuscript. Some detailed concerns are listed as below:

Response: We sincerely thank you for giving us the professional and thoughtful comments and suggestions to improve the quality of this manuscript. According to your suggestions, we have performed additional experiments and discussion in order to make following points more clearly. We hope that our response addresses your concerns.

Question 1. The manuscript mentioned that the crossed π -conjugation structure suppresses the dissolution of Ru, but the experiment data is not enough to support this argument. The ICP-OES results of after stability tested RuO₂ should be provided to illustrate it (i.e. the dissolved amount of Ru for COF-205-Ru and RuO₂ during stability test).

Response: We thank the reviewer for your careful consideration from the perspective of Ru dissolution and it is useful to improve the quality of this work. Indeed, 2,2'-bipyridine (2,2'-Bpy) unit is commonly employed in complexation due to its robust redox stability and ease of functionalization (*Chem. Rev.* 2000, **100**, 3553–3590).

Thanks to the strong Ru-N interaction and its OER inertia, the bpy unit can be employed as a suitable building block to achieve efficient and stable photocatalytic/electrocatalytic water oxidation (*Chem. Mater.* 2016, 28, 4375–4379; *Nat. Catal.* 2022, 5, 414–429; *ACS Catal.* 2020, 10, 5299–5308). As you suggested, we further assessed the catalysts' electrochemical stability during OER catalysis to better understand the relationship between OER activity and stability. According to reported method, we determined the concentration of dissolved Ru ions in the electrolytes to quantify the dissolution of catalysts by performing ICP-MS analysis at various reaction times by maintaining current density of 10 mA cm⁻² (*Adv. Mater.* 2023, 35, 2305939; *ACS Catal.* 2020, 10, 12182-12196; *Nat. Commun.* 2023, 14, 7644). As shown in **Figure R1** (**Supplementary Fig. 31** in the Revised Supplementary information), the concentration of dissolved Ru ions in RuO₂ increases significantly with increasing reaction time (6.91% after 10 h operation), while that in COF-205-Ru increases only slightly (only 0.61% after 10 h operation), indicating the introduction of Ru-N motifs hinders Ru from dissolving kinetically.

Figure R1. Ru dissolved mass fraction of COF-205-Ru and RuO₂ after operating times of 1, 3, 6, and 10 h under 0.5 M H₂SO₄ electrolyte.

Accordingly, *Figure R1* is shown in the Supplementary Information as *Supplementary Fig. 31*. And the relevant discussion have been supplemented in the Revised Manuscript as follow:

“We quantified the Ru dissolution for COF-205-Ru and commercial RuO₂ during chronopotentiometry measurement (at 10 mA cm⁻²) by ICP-MS analysis on electrolyte

aliquots taken after 1, 3, 6, and 10 h (ACS Catal. 2020, 10, 12182-12196). As shown in Supplementary Fig. 31, the Ru dissolution in RuO₂ increases significantly (6.91% after 10 h), while that in COF-205-Ru is maintained well (0.61% increase after 10 h).” (the highlighted text in the *Electrocatalytic OER evaluation* section, Page 13)

Question 2. While the electrocatalyst have a long-term durability of 125 h, I wonder did the electronic structure of COF-205-Ru change during the stability test. XAFS or XPS of after stability tested COF-205-Ru is suggested to provide to monitor the valence state change during OER. Moreover, characterizations to monitor the change of ligand are also necessary.

Response: Thanks for your constructive suggestion and it is necessary to verify whether the chemical structure of COF-205-Ru changed after OER test. We carefully analyzed high-resolution Ru 3*p* XPS spectrum of COF-205-Ru to further probe the valence states of Ru. As shown in **Figure R2** (**Supplementary Fig. 29** in Revised Supplementary Information), the XPS peaks of Ru 3*p* exhibit almost no change after OER test, unveiling its excellent chemical stability during OER process.

2). In order to monitor whether the ligand dissolves during OER, the crystal structure is surveyed by *in situ* XRD. As shown in **Figures R3 and R4** (**Supplementary Fig. 26** and **Fig. 4i** in Revised version), to our delight, compared with commercial RuO₂ reference, COF-205-Ru exhibits a much lower peak intensity loss ratio than that of RuO₂, further revealing its robust framework under OER operating potentials.

3). Based on the above discussion, the N 1*s* XPS spectra of COF-205-Ru before and after OER were collected to track the change on coordinated structure. As presented in **Figure R5** (**Supplementary Fig. 30** in Revised Supplementary Information), the XPS peak of N 1*s* is almost unaltered after OER test.

Figure R2. High-resolution XPS of Ru 3p for COF-205-Ru before and after OER.

Figure R3. *In situ* PXRD patterns of (a) COF-205-Ru and (b) commercial RuO₂ reference at different applied potentials.

Figure R4. Peak intensity loss ratio of main crystal plane for COF-205-Ru and commercial RuO₂ from *in situ* PXRD patterns.

Figure R5. High-resolution XPS of N 1s for COF-205-Ru before and after OER.

Accordingly, Figures R4 has been added in Fig. 4i, Figures R2, R3 and R5 have been supplemented in the Revised Supplementary Information as Supplementary Figs. 29, 26 and 30, respectively. And the relevant discussion has been added in the Revised Manuscript as follow:

“To our delight, compared with commercial RuO₂ reference, COF-205-Ru exhibits a much lower peak intensity loss ratio, further revealing its robust framework under OER operating potentials (Supplementary Fig. 26 and 4i).” (the highlighted text in the *Electrocatalytic OER evaluation* section, Page 13)

Question 3. The Ru 3p XPS spectra shows that the binding energy of Ru 3p_{3/2} is positively shifted compared to that of RuO₂ and Ru foil. How could this reveal that the valence state of Ru in COF-205-Ru is between 0 and +4? Related detailed discussion should be provided.

Response: We are sincerely grateful for your valuable suggestion, which has enhanced the quality of our work. In the revised manuscript, we carefully analyzed and deconvoluted the high-resolution Ru 3p XPS spectrum of COF-205-Ru and RuO₂ to further clearly probe the valence states of Ru. As shown in **Figure R6** (Fig. 3b in Revised Manuscript), the Ru 3p spectra were deconvoluted into two sets of doublets. The peaks centered at binding energies of 464.5 and 486.9 eV are attributed to Ru³⁺, and the doublet with BEs of 462.7 and 484.9 eV belong to Ru⁴⁺ (*Adv. Energy Mater.* **2019**, *9*, 1901313; *Angew. Chem. Int. Ed.* **2022**, *61*, e202200211). Moreover, the XPS peak intensity of Ru⁴⁺ decreased from RuO₂ to COF-205-Ru, demonstrating the decreased Ru valence caused by the electronic interaction between COF framework and Ru species, this trend is in consistent with above XAS results.

Figure R6. High-resolution XPS of Ru 3p for COF-205-Ru and RuO₂ benchmark.

Accordingly, Figure R6 has been added in the Revised Manuscript as Fig. 3b. And the relevant discussion have been supplemented in the Revised Manuscript as follow:

“For deconvoluted Ru 3p XPS spectra, the peaks centered at binding energies of 464.5 and 486.9 eV are attributed to Ru³⁺, and the doublet with binding energies of 462.7 and 484.9 eV belong to Ru⁴⁺ (Adv. Energy Mater. 2019, 9, 1901313; Angew. Chem. Int. Ed. 2022, 61, e202200211). Moreover, the XPS peak intensity of Ru⁴⁺ decreased from RuO₂ to COF-205-Ru, demonstrating the decreased Ru valence caused by electronic interaction between COF framework and Ru species, this trend is in consistent with above XAS results.” (the highlighted text in the *Activation of the coordinated oxygen* section, Page 9)

Question 4. How to explain that the N 1s XPS spectra of COF-205 and COF-205-Ru show an increase of Graphitic N after anchoring Ru?

Response: We sincerely apologize for the misunderstanding about the graphite N signal, and this valuable question facilitates to improve the quality of this manuscript. Considering that the formation conditions of graphite N are relatively harsh, it generally comes from some amorphous substances in the synthesis process, where facile Ru loading process will not theoretically cause the increase of graphite N content. The M-N coordination causes the peak intensity of pyridine N to decrease significantly, this may result in a visually enhanced content of graphite N. As expected, the calculated XPS peak area ratio of graphite N exhibits almost no change from COF-205 (10.2%)

and COF-205-Ru (11.0%), demonstrating the well-maintained graphite N content after anchoring Ru (**Figure R7**, **Figs. 3e** and **3f** in Revised Manuscript).

Figure R7. N 2p XPS spectra of COF-205 and COF-205-Ru to demonstrate the Ru-N interaction.

Accordingly, Figure R7 has been integrated into the Revised Manuscript as Fig. 3. And the relevant discussion have been supplemented in the Revised Manuscript as follow:

“The calculated XPS peak area ratio of graphite N exhibits almost no change from COF-205 (10.2%) and COF-205-Ru (11.0%), demonstrating the well-maintained graphite N content after anchoring Ru (Figs. 3e and 3f).” (the highlighted text in the *Activation of the coordinated oxygen* section, Page 9)

Question 5. According to the EXAFS that authors use to support their claim seems to be contradictory to the reference. In the reference (J. Am. Chem. Soc. 143, 6482-6490 (2021)), Ir-O bond length have a shrinkage, indicating the increased covalency Ir-O bond.

Response: Thanks for your professional and thoughtful consideration about the M-O covalency. In view of this problem, different explanations have been proposed in previous literatures. Herein, we believe that the elongated Ru-O bond is beneficial to enhance the covalency of Ru-O to stimulate lattice oxygen to participate in OER reaction, for the following reasons:

1). Firstly, according to *crystal field molecular orbital theory*, the elongated Ru-O bond will lead to the geometrical distortion of RuO₆ octahedron and further split the Ru *d* orbital (**Figure R8**, **Supplementary Fig. 17** in Revised Supplementary information).

As a result, the Ru *d* orbital is rearranged, which leads to the increased Ru-O covalency (Nat. Sci. Rev. **2023**, 10, nwad010; Nat. Energy **2018**, 3, 373–386; Adv. Mater. **2023**, 35, 2302462). The greater M-O covalency can facilitate intramolecular electron transfer from oxygen ligands to M cations within the lattice matrix, resulting in ligand hole formation for lattice oxygen activation (J. Am. Chem. Soc. **1995**, 117, 8557-8566; Energy Environ. Sci. **2021**, 14, 4647-4671; Nat. Catal. **2020**, 3, 554-563; Nat. Chem. **2017**, 9, 457-465; Nat. Commun. **2022**, 13, 3784; Adv. Mater. **2018**, 30, 1802912). For instance, Li and co-workers enhanced Co-O covalency by *d-p-f* orbital hybridization, with the elongated Co-O bond length (Adv. Mater. **2023**, 35, 2302462). Therefore, we believe that the elongated Ru-O bond can effectively enhance the Ru-O covalency and activate the lattice oxygen.

Figure R8. (a) Effect of ligand field variation on d-orbital energy level splitting; (b) Illustration of the larger hybridization degree between Ru-*d* orbital and O-*p* orbital.

2). On the other hand, there have also been many previous reports suggesting that Ru-O bond contraction can facilitate AEM pathway (Nat. Commun. **2019**, 10, 162; Adv. Energy Mater. **2023**, 13, 2300177). For example, Sargent et al claimed that:“ The strong interaction in these Ru–O–Ir local structure (a weak bonding of lattice oxygen: Ru–O in SrRuIr was slightly increased compared to RuO₂, while the Ir–O bonds were reduced compared to IrO₂) may suppress the lattice oxygen involvement during OER (LOM pathway), thus improving the stability of the electrocatalyst” (J. Am. Chem. Soc. **2021**, 143, 6482–6490).

3). Moreover, as we known, RuO₂ follows the LOM pathway during OER process.

Based on this, in principle, the longer Ru-O bond has a smaller bond energy, which makes it easier to break and remove from the lattice, coupling with the *O intermediate to achieve Vo-*OO_L species, that is, accelerates LOM pathway.

Accordingly, Figure R8 has been added in the Revised Supplementary information as Supplementary Fig. 17. And we have added the relevant discussion in the Revised Manuscript as follow:

*“This geometrical distortion further leads to the rearrangement of Ru d orbital, which leads to the enhanced hybridization degree between Ru-d orbital and O-p orbital (Nat. Energy **2018**, 3, 373–386; Adv. Mater. **2023**, 35, 2302462), and increased Ru-O covalency (Supplementary Fig. 17) (Joule 2021, 5, 2164-2176; J. Am. Chem. Soc. 1995, 117, 8557-8566, Energy Environ. Sci. **2021**,14, 4647-4671; Nat. Catal. **2020**, 3, 554-563; Nat. Chem. **2017**, 9, 457-465; Nat. Commun. **2022**, 13, 3784; Adv. Mater. **2018**, 30, 1802912). (the highlighted text in the **Activation of the coordinated oxygen** section, Page 10)*

Question 6. The order of Fig.2g-2i do not correspond to the manuscript (L162-169).

Response: Thanks for your reminding. As shown in **Figure R1** (Fig. 2 in Revised Manuscript), we have corrected the order of Fig. 2g-2i to match the main text accordingly in the Revised Manuscript.

Fig. R9 | The structural characterization of the as-prepared COF-205 and COF-205-Ru. **a**, 2D SAXS images and Pawley refinement of experimental SAXS data for the as-synthesized COF-205, where blue circles represent experimental data; red lines represent calculated data; green lines show the difference and purple bars show the Bragg position. **b**, Experimental PXRD patterns of COF-205 and COF-205-Ru. **c**, Simulated AA and AB stacking model for COF-205. **d**, FT-IR spectra of COF-205, COF-205-Ru and corresponding monomers. **e**, Raman spectra of COF-205 and COF-205-Ru. **f**, ^{13}C CP-MAS NMR signals of COF-205 and COF-205-Ru. **g**, The pH durability evaluation for COF-205-Ru under 12 M HCl and 14 M NaOH; **h,i**, Nitrogen isotherm of COF-205 (**h**) and COF-205-Ru (**i**) at 77 K. Filled and open circles represent adsorption and desorption stages, respectively. Inset: corresponding pore size distribution. **j-l**, TEM (**j**), HRTEM (**k**) and TEM-EDS (**l**) images of COF-205-Ru. Scale bar: 100 nm for (**j**), 5 nm for (**k**) and 200 nm for (**l**), respectively.

Again, we sincerely appreciate your valuable efforts and professional suggestions

to improve the quality of this work.

Reviewer #3:

In this manuscript, the authors have successfully developed a robust conjugated COF framework to encapsulate Ru sites for acidic OER electrocatalysis, which demonstrated simultaneously improved stability and activity compared to the commercial RuO₂. Interestingly, the COF-205-Ru can not only facilitate the LOM pathway by activating the lattice oxygen and stabilizing the oxygen vacancies, but also maintain the structural stability by robust frameworks and Ru-N motifs. Different with conventional elemental-doping engineering, this work provides a new direction to develop efficient OER catalysts. Overall speaking, the work is well organized and written, the results would be enlightening. As a result, it could be published in this journal after addressing some questions below.

Response: We sincerely appreciate the reviewer for your positive evaluation of this work. And we have revised the manuscript carefully and supplemented the relevant discussion.

Question 1. Please provide more details besides XPS to further prove the Ru-N interaction.

Response: Thank you for your valuable comment and it is useful to improve the quality of this work. In principle, 2,2'-bipyridine (2,2'-Bpy) unit is commonly employed in complexation due to its robust redox stability and ease of functionalization (*Chem. Rev.* **2000**, *100*, 3553–3590; *Angew. Chem. Int. Ed.* **2023**, *62*, e202217527). Thus, it is reasonable to embed Ru ions into this 2,2'-bipyridine-containing 2D COF by Ru-N interaction. As you mentioned, as depicted in **Figure R1** (Figs. 3e-3f in Revised Manuscript), by analyzing the N 2p spectra, an additional peak located at about 399.5 eV can be observed in COF-205-Ru (*Adv. Funct. Mater.* **31**, 2107072 (2021); *Nat. Commun.* **13**, 5843 (2022)), further confirming the strong Ru-N interaction. In addition, the structural parameters extracted from the Ru K-edge EXAFS fitting also confirm these results.

As you suggested, we further supplement synchrotron radiation soft X-ray absorption near-edge spectroscopy (XANES) measurement to investigate the atomic coordination structure of Ru ions and COF-205 frameworks. As displayed in **Figure R2** (**Supplementary Fig. 14** in Revised Supplementary Information), a well-defined spectroscopic peak (N1, ~399 eV) observed in the N K-edge XANES spectra of COF-205 can be assigned to the pyridinic N 1s $\rightarrow \pi^*$ resonance (*Angew. Chem. Int. Ed.* **2022**, *61*, e202210789; *Adv. Sci.* **2023**, *10*, 2302623). The peak N2 represents general transitions from the N 1s core level to C-N σ^* states, not the characteristic signal of any specific N species (*Angew. Chem. Int. Ed.* **2021**, *60*, 25296–25301; *J. Phys. Chem. C* **2014**, *118*, 7765–7771). It is worth noting that, compared to COF-205, an additional peak located at approximately 401 eV for COF-205-Ru could be observed, which can be assigned to Ru–N interaction (*Adv. Mater.* **2021**, *33*, 2007508; *Nat. Commun.* **2022**, *13*, 5843; *Angew. Chem. Int. Ed.* **2023**, *62*, e202308775), indicating the successful formation of Ru–N bonds.

Figure R1. N 2p XPS spectra of COF-205 and COF-205-Ru to demonstrate the Ru–N interaction.

Figure R2. N K-edge XANES spectra of COF-205 and COF-205-Ru.

Accordingly, Figure R2 is shown in the Supplementary Information as Supplementary Fig. 14. The relevant experimental details are supplemented in the Revised Manuscript as follow:

“Moreover, we further supplement synchrotron radiation soft X-ray absorption near-edge spectroscopy (XANES) measurement to investigate the atomic coordination structure of Ru and COF-205 framework. As displayed in Supplementary Fig. 14, a well-defined spectroscopic peak (N1, ~399 eV) observed in the N K-edge XANES spectra of COF-205 can be assigned to the pyridinic N 1s → π resonance (Angew. Chem. Int. Ed. **2022**, 61, e202210789; Adv. Sci. **2023**, 10, 2302623). The peak N2 represents general transitions from N 1s core level to C-N σ* states (Angew. Chem. Int. Ed. **2021**, 60, 25296–25301; J. Phys. Chem. C **2014**, 118, 7765–7771). It is worth noting that, compared to COF-205, an additional peak located at approximately 401 eV for COF-205-Ru could be observed, which can be assigned to Ru–N interaction, indicating the successful formation of Ru-N bonds.”* (the highlighted text in the *Activation of the coordinated oxygen* section, Page 9)

Question 2. To get more insight into the reaction kinetics mechanism, the authors should supplement KIE investigation by H/D isotope-labelled experiment.

Response: We are sincerely grateful for your valuable suggestion. To date, the use of KIE is an established experimental technique to study chemical reactions involving proton-involving transfer and/or oxygen evolving (ACS Catal. **2022**, 12, 5345; Science **2011**, 334, 1383; ACS Catal. **2018**, 8, 816; J. Phys. Chem. Lett. **2011**, 2, 2200). As discussed in *Question 4*, both COF-205-Ru and RuO₂ display strong pH-dependent OER kinetics, which indicates that these electrocatalysts proceed via a non-concerted proton–electron transfer (NCPET) process (Nat. Chem. **2017**, 9, 457–465; Catal. Today **2016**, 262, 2–10). Accordingly, proton transfer process may not be involved in the rate-determining step (RDS), therefore, the kinetic isotope effects (KIE) investigation by H/D isotope-labelled experiment should be employed to reflect the proton transfer

kinetic information and determine the reaction RDS (*J. Am. Chem. Soc.* **2007**, *129*, 5870–5879; *Mater.* **2013**, *6*, 392–409; *J. Am. Chem. Soc.* **2004**, *126*, 9786–9795; *Proc. Natl Acad. Sci. USA* **2010**, *107*, 7225). As shown in **Figure R3** (Supplementary Fig. 35 in Revised Supplementary information), we surveyed the electrocatalytic performance of COF-205-Ru in 0.5 M H₂SO₄ dissolved in H₂O or D₂O. The use of D₂O decreases the OER activity of COF-205-Ru. The KIE value could be calculated based on the reaction current density in the protonic vs. deuterium solution at the same overpotential. The presence of KIEs (KIEs > 1.5) is considered as evidence that proton transfer is involved in the RDS (*Nat. Commun.* **2019**, *10*, 5074; *Nat. Commun.* **2021**, *12*, 3036; *J. Am. Chem. Soc.* **2019**, *141*, 2938–2948; *ACS Catal.* **2017**, *7*, 2770–2779). However, COF-205-Ru displays the KIE values in OER potential regions fluctuate around the upper limit of secondary KIE (~1.5) with the absence of primary KIE, indicating that proton transfer is not RDS for the acidic OER on COF-205-Ru catalysts. Combined with the molecular probe (TMA⁺) analysis, we successfully recognized the peroxo-like species from the coordinated oxygen involved pathway, which is consistent with DFT calculations.

Figure R3. (a) Polarization curves of COF-205-Ru in 0.5 M H₂SO₄ dissolved in H₂O or D₂O. (b) The corresponding KIE values at different overpotentials.

Accordingly, Figure R3 is shown in the Supplementary Information as Supplementary Fig. 35. The relevant discussion has been added in the Revised Manuscript as follow:

“Note that, for NCPET process, the proton transfer process may not be involved

*in rate-limiting step, therefore, the kinetic isotope effects (KIE) investigation by H/D isotope-labelled experiment should be employed to reflect the proton transfer kinetic information and determine the reaction RDS (J. Am. Chem. Soc. 2007, 129, 5870–5879; Materials 2013, 6, 392-409; J. Am. Chem. Soc. 2004, 126, 9786–9795; Natl Acad. Sci. USA 2010, 107, 7225-7229). As shown in Supplementary Fig. 35a, the use of D₂O decreases the activity of COF-205-Ru. The KIE values are calculated based on the reaction current density in the protonic vs. deuterium solution at the same overpotential (Supplementary Fig. 35b). The presence of KIEs (KIEs > 1.5) is considered as evidence that proton transfer is generally involved in the RDS (Nat. Commun. 2019, 10, 5074; Nat. Commun. 2021, 12, 3036; J. Am. Chem. Soc. 2019, 141, 2938–2948; ACS Catal. 2017, 7, 2770–2779), while COF-205-Ru exhibits KIE values less than secondary KIE (~1.5), indicating that proton transfer is not RDS.” (the highlighted text in the **Electrocatalytic OER evaluation** section, Page 14)*

Question 3. I am interested in conjugated sp² carbon-linked COF-205-Ru to achieve efficient and stable acidic OER. In this view, stability test at large current densities (such as 50 mA cm⁻²) or long-term V-T test (more than 200 h) should be added to further support the stable feature of the catalysts.

Response: Thanks for your constructive suggestion and it is useful to further improve the quality of this manuscript. Developing efficient electrocatalysts for OER at large-current-density or last long-term in acidic media is appealing and challenging for large-scale water electrolysis. Thus, it is reasonable to consider the harsh electrocatalytic stability from the perspective of robust sp² carbon-linked structure. As your suggestion, we supplemented the chronopotentiometric measurement to survey the electrochemical stability of COF-205-Ru under 0.5 M H₂SO₄ electrolyte. As shown in **Figure R4 (Fig. 4h** in Revised Manuscript), COF-205-Ru can operate stably for over 280 h under 0.5 M H₂SO₄ electrolyte at current density of 10 mA cm⁻². In addition, COF-205-Ru exhibits nearly unaltered OER performance over a period of 100 h at 50 mA cm⁻², representing excellent acidic OER stability (**Figures R5, Supplementary Fig. 24** in Revised Supplementary Information).

Figure R4. Chronopotentiometry measurements of COF-205-Ru at current density of 10 mA cm^{-2} .

Figure R5. Chronopotentiometry measurements of COF-205-Ru at current density of 50 mA cm^{-2} .

Accordingly, Figure R4 is shown in the Revised Manuscript as Fig. 4h, Figure R5 is shown in the Revised Supplementary information as Supplementary Fig. 24. The relevant discussion has been supplemented in the Revised Manuscript as follow:

“Furthermore, COF-205-Ru exhibits nearly unaltered OER performance during the chronopotentiometric test over a period of 280 h at 10 mA cm^{-2} , which is much better than the commercial RuO_2 (10 h) (Fig. 4h).” (the highlighted text in the *Electrocatalytic OER evaluation* section, Page 13)

“Interestingly, the COF-205-Ru can operate stably for over 100 h under $0.5 \text{ M H}_2\text{SO}_4$ electrolyte even at current density of 50 mA cm^{-2} , representing excellent acidic OER stability (Supplementary Fig. 24).” (the highlighted text in the *Electrocatalytic OER evaluation* section, Page 13)

Question 4. For pH-dependence experiments, the different solution resistance (R_s) for these H_2SO_4 electrolytes have to considered.

Response: Thanks for your constructive suggestion. As you suggested, we have added

the pH-dependent activity after iR -correction in the Revised Manuscript. We simply vary the electrolyte pH between 0 and 1 with H_2SO_4 , and the iR -corrected LSVs of COF-205-Ru and RuO_2 in different electrolytes at a scan rate of 10 mV s^{-1} are shown in **Figure R6** (Figs. 5a-5b in the Revised Manuscript) and **Figure R7** (Supplementary Fig. 33 in the Revised Supplementary Information). As expected, COF-205-Ru and RuO_2 all show strong pH-dependence. The catalytic activity of COF-205-Ru and RuO_2 with pH-dependence indicates a non-concerted proton–electron transfer (NCPET) process (*Nat. Chem.* **2017**, *9*, 457-465; *Catal. Today* **2016**, *262*, 2-10). In order to visually show the strength of pH-dependence, we compared the current densities of the above three catalysts at different pH values at 1.55 V vs. RHE and 1.60 V vs. RHE, respectively, and the results are shown in **Figure R7b**. Besides, good linear relationships are observed between the corresponding $\log(j)$ at different potentials with electrolyte pH values.

Figure R6. The pH-dependence of catalytic performance on COF-205-Ru. (a) iR -corrected LSVs of COF-205-Ru in H_2SO_4 electrolytes (pH 0–1) at a scan rate of 10 mV s^{-1} ; (b) corresponding $\log(j)$ at 1.48 V and 1.58 V under different pH values.

Figure R7. The pH-dependence of catalytic performance on commercial RuO_2 . (a) iR -corrected

LSVs of commercial RuO₂ in H₂SO₄ electrolytes (pH 0–1) at a scan rate of 10 mV s⁻¹; (b) corresponding log(*j*) at 1.48 V and 1.58 V under different pH values.

Accordingly, Figure R6 has been added in the Revised Manuscript as Fig. 5a-5b, Figure R7 has been added in the Revised Supplementary Information as Supplementary Fig. 33. And the relevant discussion has been added in the Revised Manuscript as follow:

“We acquired the *iR*-corrected linear sweep voltammetry (LSV) curves of COF-205-Ru in H₂SO₄ (pH 0–1) at a scan rate of 10 mV s⁻¹ (Fig. 5a), with commercial RuO₂ (recognized LOM pathway) for comparison (Supplementary Fig. 33).” (the highlighted text in the *Electrocatalytic OER evaluation* section, Page 14)

Question 5. After the stability test, the authors use in situ XRD, TEM and ICP to illustrate the well-maintained phase structure, morphology and metal content, I think it is necessary to supplement characterization to verify whether the metal valence of COF-205-Ru changed after OER test.

Response: Thanks for your constructive suggestion and it is necessary to verify whether the metal valence of COF-205-Ru changed after OER test. We carefully analyzed high-resolution Ru 3*p* and N 1*s* XPS spectrum of COF-205-Ru to further probe the valence states of Ru and N. As shown in Figure R8 (Supplementary Figs. 29 and 30 in Revised Supplementary Information), the XPS peaks of Ru 3*p* exhibit almost no change after OER test, unveiling its excellent chemical stability during OER process. Similarly, the XPS peak of N 1*s* also presents unaltered after OER test, which is in consistent with above results.

Figure R8. High-resolution XPS of (a) Ru 3p and (b) N 1s for COF-205-Ru before and after OER.

Accordingly, Figure R8 has been added in the Revised Supplementary Information as Supplementary Figs. 29 and 30. And the relevant discussion has been added in the Revised Manuscript as follow:

“In addition, the morphology, Ru content, valence state and coordinated structure for COF-205-Ru after OER cycling test are maintained well, indicating the excellent durability (Supplementary Figs. 27-30).” (the highlighted text in the *Electrocatalytic OER evaluation* section, Page 13)

Question 6. The authors are required to provide turnover frequency (TOF) values of COF-205-Ru and other samples to evaluate catalytic performance.

Response: We are sincerely grateful for your constructive comment. And as you suggested, we have given the details of TOF calculation in the Revised Manuscript.

The TOF values of the as-prepared samples, representing the activity of a single site, were calculated. For OER, the TOF value is calculated by the equation (*Adv. Mater.* **2023**, *35*, 2305659; *Energy Environ. Sci.* **2021**, *14*, 6428-6440):

$$\text{TOF} = (j \times A) / (4 \times F \times n)$$

where j (mA cm⁻²) is the current density at an applied potential; A is the geometric area of the GCE (0.196 cm²), F is Faraday's constant (86485 C mol⁻¹) and n is the molar number of active sites. In this work, the number n (mol) was estimated with the following equation:

$$n = (m \times N_A) / M_w$$

where m is the loading mass of Ru, N_A and M_w are the Avogadro's constant and the molecular weight, respectively.

As shown in **Figure R9** (Supplementary Figs. 23 in Revised Supplementary information), the TOF values of COF-205-Ru and commercial RuO₂ reference were calculated based on the loading amounts of Ru to investigate the intrinsic activity of

active site. Remarkably, the TOF value of COF-205-Ru is superior than commercial RuO₂. Specifically, the TOF value is calculated to be 0.47 s⁻¹ for COF-205-Ru at η = 300 mV, much higher than that of RuO₂ (0.014 s⁻¹), further indicating the superior OER kinetics of COF-205-Ru (*Chem. Sci.*, **2018**, *9*, 3470-3476; *Adv. Energy Mater.*, **2015**, *5*, 1500412).

Figure R9. Turnover frequency (TOF) values of COF-205-Ru and commercial RuO₂ reference.

Accordingly, Figure R8 has been added in the Revised Supporting Information as Supplementary Figs. 23. The relevant discussion has been supplemented in the Revised Manuscript as follow:

“Besides, the TOF values of COF-205-Ru and commercial RuO₂ were calculated based on the loading amounts of Ru to investigate the intrinsic activity (Supplementary Figs. 23). Specifically, the TOF value of COF-205-Ru is calculated to be 0.47 s⁻¹ at η = 300 mV, much higher than that of RuO₂ (0.014 s⁻¹).” (the highlighted text in the **Electrocatalytic OER evaluation** section, Page 12)

The relevant discussion has been supplemented in the Revised Manuscript as follow:

“The TOF values of the as-prepared samples, representing the activity of a single site, were calculated. For OER, the TOF value is calculated by the equation:

$$TOF = (j \times A) / (4 \times F \times n)$$

where j (mA cm⁻²) is the current density at an applied potential; A is the geometric area of the GCE (0.196 cm²), F is Faraday’s constant (86485 C mol⁻¹) and n is the

molar number of active sites. In this work, the number n (mol) was estimated with the following equation:

$$n = (m \times N_A) / M_w$$

where m is the loading mass of Ru, N_A and M_w are the Avogadro's constant and the molecular weight, respectively." (the highlighted text in the *Electrochemical characterizations* section, Page S4)

Question 7. As shown in Figure 4c, the authors have compared the activity with other recently reported acidic OER catalysts. However, the displayed overpotential of COF-205-Ru is about 320 mV, please correct it.

Response: Thanks for your careful inspection and we sincerely apologize for making such mistake. As you suggested, we have corrected this error in the text accordingly in the Revised Manuscript (**Figure R10**, **Fig. 4c** in Revised Manuscript).

Figure R10. Comparison of the mass activity for COF-205-Ru with typically reported OER catalysts under acidic solution.

Question 8. *In situ* XRD should use attenuation ratio of characteristic diffraction peak rather than stack lines. As a comparison, the *in situ* XRD data of commercial RuO₂ deserve to be supplemented.

Response: Thanks for your professional and thoughtful consideration. Indeed, *in situ* PXRD patterns can provide the in-depth information and realistically assess the durability of COF-205-Ru at different applied potentials (*Angew. Chem. Int. Ed.*

2023, 62, e202313886). The attenuation ratio of characteristic diffraction peak is more favorable to compare the chemical stability. As you suggested, we supplemented the *in situ* XRD data of commercial RuO₂ as a comparison. To our delight, compared with commercial RuO₂ reference, COF-205-Ru exhibits a much lower peak intensity loss ratio than that of RuO₂, further revealing its electrochemical stability under OER operating potentials (**Figures R11 and R12**).

Figure R11. *In situ* PXRD patterns of (a) COF-205-Ru and (b) commercial RuO₂ reference at different applied potentials.

Figure R12. Peak intensity loss ratio of main crystal plane for COF-205-Ru and commercial RuO₂ from *in situ* PXRD patterns.

Accordingly, Figure R11 has been added in the Revised Supporting Information as Supplementary Fig. 26. Figure R11 has been added in the Revised Manuscript as Fig. 4i. The relevant discussion has been supplemented in the Revised Manuscript as follow:

“To our delight, compared with commercial RuO₂, COF-205-Ru exhibits a much lower peak intensity loss ratio than that of RuO₂, further revealing its robust framework

under OER operating potentials (Fig. 4i and Supplementary Fig. 26).” (the highlighted text in the *Electrocatalytic OER evaluation* section, Page 13)

REVIEWER COMMENTS

Reviewer #1 (Remarks to the Author):

Authors did extensive revisions and addressed most of my comments. I am still very concerned with the use of carbon supports (glassy carbon and carbon cloth) in OER reaction. We all know that carbon is never used in electrolyser and it corrodes under oxidizing conditions. Authors need to determine if any CO₂ is produced and Faradic efficiency from H₂O to O₂ (parasitic process can be CO₂ production). Or they should report activity on more typical electrolyser support stable under oxidizing conditions such as Ti mesh, FTO or ITO.

Reviewer #2 (Remarks to the Author):

The authors did new suits of experiments and studies. The concerns have been addressed. I can recommend it to be published in the present version.

Reviewer #3 (Remarks to the Author):

In the revised manuscript, I think the authors have revised all the points according to the review comments and I think it can be published without further revision.

Response Letter

Reviewer #1:

Authors did extensive revisions and addressed most of my comments.

Response: We sincerely appreciate you for your positive evaluation of the revised manuscript.

Question 1. I am still very concerned with the use of carbon supports (glassy carbon and carbon cloth) in OER reaction. We all know that carbon is never used in electrolyser and it corrodes under oxidizing conditions. Authors need to determine if any CO₂ is produced and Faradic efficiency from H₂O to O₂ (parasitic process can be CO₂ production). Or they should report activity on more typical electrolyser support stable under oxidizing conditions such as Ti mesh, FTO or ITO.

Response: Thank you for your specialized question and valuable suggestions, this is helpful to improve the quality of this work. We agree with you that carbon electrodes inevitably corrode at OER potentials, therefore, it is necessary to determine whether it interferes with the assessment of OER activity. Herein, we have supplemented the online differential electrochemical mass spectrometry (DEMS), Faradic efficiency and OER test on Ti mesh to investigate this question. The results show that although there is a slight corrosion of the carbon electrode at the OER potential, the current contribution is quite small and does not interfere with our assessment of OER activity. The specific responses are as follows:

(1) As you suggested, to determine if any CO₂ is produced, we employed online differential electrochemical mass spectrometry (DEMS) to identify the gas type during OER process. Specifically, we monitored the ionic current of O₂ ($m/z = 32$) and CO₂ ($m/z = 44$) at different applied potentials (OCV, 1.50 V~ 1.75 V) for COF-205-Ru loaded on Carbon Cloth electrode, and each potential was applied for 300 s. As shown in **Figure R1**, when the applied potential over 1.55 V, a low CO₂ ionic current was

detected. Indeed, this can be attributed to the slight corrosion of CC electrode, as you claimed. However, it can be clearly seen from **Figure R1** that the ionic current of CO₂ is much lower than that of O₂. Specifically, at 1.75 V (vs. RHE), the ionic current of O₂ (8.5×10^{-11} A) is more than 35 times enhancement compared to CO₂ (2.4×10^{-12} A), demonstrating that the OER current basically comes from the O₂ evolution rather than the oxidation of carbon electrode.

(2) In addition, the faradaic efficiency (FE) for COF-205-Ru is nearly 100% (**Figure R2**), as determined by comparing the experimentally obtained oxygen volume with the theoretical expectation, thus excluding significant contributions from side reactions.

(3) Moreover, we further evaluated the electrochemical activity and stability of COF-205-Ru and commercial RuO₂ on Ti mesh electrode. As shown in **Figure R3**, LSV curves show that the COF-205-Ru presented an excellent OER activity on Ti mesh electrode, with an overpotential of 214 mV at 10 mA cm^{-2} , which is in accordance with GCE electrode, demonstrating its excellent intrinsic OER kinetics. And the COF-205-Ru can operate stably for over 100 h under 0.5 M H₂SO₄ electrolyte on Ti mesh electrode at 10 mA cm^{-2} , representing its acidic OER stability.

Combining these above results, we believe that the O₂ evolution concurrently with the OER Faradic current in 0.5 M H₂SO₄ electrolyte and the COF-205-Ru has an excellent reaction kinetics toward acidic water oxidation.

Figure R1. Ionic current of O₂ ($m/z = 32$) and CO₂ ($m/z = 44$) at different applied potentials.

Figure R2. The Faradaic efficiency of COF-205-Ru under 0.5 M H₂SO₄.

Figure R3. OER performance of COF-205-Ru on Ti mesh electrode under 0.5 M H₂SO₄ electrolyte.

Accordingly, **Figures R2** and **R3** has been supplemented into Revised Supplementary Information as **Supplementary Figs. 25** and **20** and , respectively. The relevant discussions have been supplemented in the Revised Manuscript as follow:

“In addition, the faradaic efficiency (FE) for COF-205-Ru is nearly 100% (Supplementary Fig. 25), as determined by comparing the experimentally obtained oxygen volume with the theoretical expectation.” (the highlighted text in the **Electrocatalytic OER evaluation** section, Page 12)

“Similarly, the COF-205-Ru also exhibits excellent OER performance on carbon cloth or Ti mesh electrode (Supplementary Figs. 19 and 20).” (the highlighted text in the **Electrocatalytic OER evaluation** section, Page 11)

Once again, we sincerely thank this reviewer for your professional advice and we

hope our response is suitable and acceptable to solve your concerns.

REVIEWERS' COMMENTS

Reviewer #1 (Remarks to the Author):

Authors addressed all my questions and paper can be recommended for publication.